# Transfer Learning on Edge Connecting Probability Estimation Under Graphon Model

**Yuyao Wang**
Boston University
yuyaow@bu.edu

**Yu-Hung Cheng**
Boston University
yhcheng@bu.edu

**Debarghya Mukherjee**
Boston University
mdeb@bu.edu

**Huimin Cheng**
Boston University
huimin23@bu.edu

## Abstract

Graphon models provide a flexible nonparametric framework for estimating latent connectivity probabilities in networks, enabling a range of downstream applications such as link prediction and data augmentation. However, accurate graphon estimation typically requires a large graph, whereas in practice, one often only observes a small-sized network. One approach to addressing this issue is to adopt a transfer learning framework, which aims to improve estimation in a small target graph by leveraging structural information from a larger, related source graph. In this paper, we propose a novel method, namely GTRANS, a transfer learning framework that integrates neighborhood smoothing and Gromov-Wasserstein optimal transport to align and transfer structural patterns between graphs. To prevent negative transfer, GTRANS includes an adaptive debiasing mechanism that identifies and corrects for target-specific deviations via residual smoothing. We provide theoretical guarantees on the stability of the estimated alignment matrix and demonstrate the effectiveness of GTRANS in improving the accuracy of target graph estimation through extensive synthetic and real data experiments. These improvements translate directly to enhanced performance in downstream applications, such as the graph classification task and the link prediction task.

## 1 Introduction

Graph data are increasingly common in numerous applications, from social networks [54, 14, 17] to biological systems [5] and power electronics networks [64]. A fundamental task in graph learning is estimating the probability of connections between nodes, facilitating various downstream analyses. Traditional methods for estimating connection probabilities often rely on specific parametric random graph models, including Erdős–Rényi (ER) model [20, 18], the stochastic block model (SBM) [24] and SBM variants [27, 31, 8], the exponential random graph model (ERGM) [35], and latent position model [23]. While useful, these can be restrictive and suffer from model misspecification.

**Graphon.** To address this limitation, the graphon model provides a powerful non-parametric framework [34, 42]. A graphon is a symmetric, measurable function $f : [0,1]^2 \to [0,1]$, where nodes are assigned latent positions $u_i \sim \text{Unif}[0,1]$, and edge probabilities are given by $p_{ij} = f(u_i, u_j)$. This model includes classical models as special cases—e.g., constant functions yield ER graph, and step functions yield SBMs. Graphon models have been widely used for various downstream applications, such as data augmentation [22], link prediction [67], causal inference under network interference [32], dataset distillation [61], and assessing vulnerabilities in the smart grid [3]. In the existing literature, various methods have been developed for estimating the connection probability under graphon model [10, 1, 9, 11, 65, 45]. The estimation accuracy of these methods typically improves as the network size (i.e., number of nodes) increases.

39th Conference on Neural Information Processing Systems (NeurIPS 2025).

**Motivating example.** Small graphs severely limit graphon estimation accuracy, posing significant challenges for graphon-based downstream analysis. For example, in the PROTEINS-Full dataset ([22]), each graph represents a protein structure and consists of only 25 nodes on average. Therefore, employing standard graphon estimation strategies may lead to poor accuracy. Fortunately, similar domains often offer larger graphs with related structures, e.g., the D&D dataset averaging 284 nodes per protein graph. Both datasets encode similar biological relationships, creating an ideal transfer learning opportunity. Therefore, it would be highly beneficial to develop a statistically sound methodology for *transferring knowledge* from a related large graph. Such an approach could substantially enhance inference on a smaller graph, while preserving the flexibility of the graphon framework.

**Transfer learning.** Transfer learning has emerged as a powerful framework for leveraging knowledge from data-rich domains to improve learning in data-scarce domains [43]. In graph settings, it has been used for GNN meta-learning, pre-training/fine-tuning and adversarial adaptation, as well as for network regression [63, 25, 47, 7, 62, 12]. To the best of our knowledge, there is only one work [26] proposing a transfer learning method for graphon estimation. However, their methodology is constrained to scenarios where the target network exists as a subset of the source network, thereby leveraging known node correspondences between networks. We aim to tackle the more practical scenario where node correspondences are unknown. To our knowledge, no previous research addresses this gap, which presents two key challenges: (1) Alignment problem: Without known node correspondences, structural patterns may transfer to non-corresponding regions, potentially degrading estimation quality. (2) Unsupervised learning setting: Traditional transfer learning relies on labeled data, but graphon estimation provides no such signals for formulating clear transfer loss functions.

**Transfer learning for graphon estimation.** To address these challenges, we propose a novel transfer learning method for graphon estimation via optimal transport and neighborhood smoothing, abbreviated as GTRANS. Our method consists of three key steps. **Initial estimation step:** Given source and target adjacency matrices $\mathbf{A}_s \in \{0,1\}^{n_s \times n_s}$ and $\mathbf{A}_t \in \{0,1\}^{n_t \times n_t}$, $n_s > n_t$, we first apply neighborhood smoothing to obtain initial graphon estimators $\hat{\mathbf{P}}_s^{ini}$ and $\hat{\mathbf{P}}_t^{ini}$ for both source and target. **Transferring step:** We then employ Gromov-Wasserstein (GW) optimal transport to compute the alignment matrix $\hat{\pi} \in [0,1]^{n_s \times n_t}$ using the aforementioned initial graphon estimators rather than the original adjacency matrices which contain Bernoulli noise. The resulting alignment matrix is then used to map the initial source graphon estimator into the target domain's latent space through a structure-preserving projection. **Debiasing step:** When the source and target graphons differ significantly (indicated by a large GW distance in the previous step), we implement an adaptive debiasing mechanism to mitigate potential negative transfer.

Our contributions can be summarized as follows: (1) To the best of our knowledge, GTRANS is the first method for graphon estimation that transfers knowledge across graphs without any known node correspondence. (2) We establish consistency results for the proposed method, showing that the alignment matrix computed using smoothed graphon estimates converges (under mild conditions) to the true optimal transport map. (3) Extensive experiments on synthetic datasets show our method consistently achieves lower estimation error than state-of-the-art alternatives. On real-world networks, it outperforms existing approaches in both graph classification via data augmentation and link prediction tasks. Our implementation is publicly available at https://github.com/olivia3395/GTRANS.

## 2 Preliminary

In this paper, we let $\mathbf{A} \in \{0,1\}^{n \times n}$ denote an adjacency matrix, and $\mathbf{P}$ denote the corresponding probability matrix that generates $\mathbf{A}$, i.e., $\mathbf{A}_{ij} \sim \text{Ber}(\mathbf{P}_{ij})$. A graphon model typically assumes that each node $i$ has a latent position $u_i \in [0,1]$ and $\mathbf{P}_{ij} = f(u_i, u_j)$, where $f$ is a symmetric, measurable function on $[0,1]^2$. Graphon estimation refers to the estimation of the probability matrix $\mathbf{P}$ under the graphon model. The accuracy of the graphon estimation is typically measured using the mean squared error: $\text{MSE}(\hat{\mathbf{P}}, \mathbf{P}) = \|\mathbf{P} - \hat{\mathbf{P}}\|_F^2 / n^2$. MSE is widely used as a standard evaluation metric in existing literature [65, 45, 9].

### 2.1 Graphon Estimation

Broadly speaking, graphon estimation methods fall into three categories: (1) Global low-rank techniques, including Universal Singular Value Thresholding [11, 60]; (2) Combinatorial optimization, which directly searches for node assignments [19, 29] (3) Smoothing-based methods, which pool

nodes with similar degree [9] or similar latent neighborhoods [65, 40]. Among these methods, neighborhood smoothing (NS) [65] stands out because it can achieve nearly optimal MSE among all estimators computable in polynomial time. The key idea of NS method is to estimate pairwise connection probabilities by measuring the proportion of edges between the respective neighborhoods of node pairs, effectively leveraging local structure to infer global patterns. Further algorithmic details of NS are provided in Appendix G.2.

## 2.2 Gromov-Wasserstein Distance

The Gromov-Wasserstein (GW) distance [37, 38] provides a principled framework for comparing metric-measure spaces, making it particularly suitable for comparing graphs of different sizes without requiring node correspondences. GW distance and couplings (and their variations, e.g., fused GW distance [50, 41], sliced GW distance [51]) have proven to be very useful for comparing/aligning graphs, shapes, or distributions supported on different domains [15, 2, 58, 21, 36, 56]. In its most general formulation, given two cost/distance matrices $C \in \mathbb{R}^{m \times m}$ and $D \in \mathbb{R}^{n \times n}$, the GW distance between these two distance matrices is defined as: $\min_{\pi \in \Pi(\mu, \nu)} \sum_{i,j,k,\ell} L\big(C(i,k), D(j,l)\big) \pi_{ij} \pi_{kl}$ where $\Pi(\mu, \nu)$ is the set of all couplings $\pi \in \mathbb{R}^{m \times n}$ such that $\pi_{ij} \geq 0$, $\sum_{ij} \pi_{ij} = 1$ and $\sum_j \pi_{ij} = \mu_i$ and $\sum_i \pi_{ij} = \nu_j$, and $L$ is the loss function, for example $L(x, y) = (x - y)^2$. Here, $\mu$ and $\nu$ denote the uniform measures on the source and target nodes, respectively; that is, $\mu = (1/n_S, \ldots, 1/n_S)$ and $\nu = (1/n_T, \ldots, 1/n_T)$. The minimizer $\pi^*$ of the above equation is called an optimal GW coupling. While the GW distance offers a powerful framework, the exact computation of the GW distance is a quadratic assignment problem, which is known to be computationally intensive [37, 52]. The GW distance has been widely used for graph alignment and transfer tasks, demonstrating its effectiveness in applications such as node embedding, cross-domain alignment, and subgraph matching [58, 56, 13, 30, 59].

**Entropic gromov-wasserstein distance for scaling up to large datasets.** To alleviate the computational difficulty, [49] proposed adding an entropic regularization to the GW objective function as follows: $\min_{\pi \in \Pi(\mu, \nu)} \sum_{i,j,k,\ell} L\big(C(i,k), D(j,l)\big) \pi_{ij} \pi_{kl} + \epsilon \mathsf{KL}\left(\pi \mid \mu \otimes \nu\right)$. The product $\mu \otimes \nu$ is the product measure assigning a uniform weight of $(n_S n_T)^{-1}$. Although the addition of an entropic regularizer does not make the optimization problem convex (unless $\epsilon$ exceeds a certain threshold), it significantly enhances computational efficiency. As elaborated in [49], this regularization enables a Sinkhorn-like algorithm, adapted from [16] for the standard optimal transport problem, which is simple to implement and typically exhibits fast convergence [46].

## 3 Transfer Learning on Connecting Probability Estimation for Graph

In this section, we first formalize the problem setup for transfer learning on connecting probability estimation, then present our method GTRANS in detail.

### 3.1 Problem Setup

We consider the problem of estimating the graphon for a target graph of relatively small size, leveraging information from a larger source graph that shares structural similarities. Formally, we have a source graph with adjacency matrix $\mathbf{A}_s \in \{0, 1\}^{n_s \times n_s}$, where $n_s = |V_s|$ is the number of nodes, and a target graph with adjacency matrix $\mathbf{A}_t \in \{0, 1\}^{n_t \times n_t}$, where $n_t = |V_t|$ is the number of nodes. We assume $n_s > n_t$, i.e., the source graph is larger than the target graph. We assume these networks are generated by the following graphon models.

$$\mathbf{A}_{s,ij} \sim \text{Ber}(\mathbf{P}_{s,ij}), \text{where } \mathbf{P}_{s,ij} = f_s(u_{s,i}, u_{s,j}), u_{s,i} \in [0, 1], u_{s,j} \in [0, 1], \quad 1 \leq i < j \leq n_s.$$

$$\mathbf{A}_{t,ij} \sim \text{Ber}(\mathbf{P}_{t,ij}), \text{where } \mathbf{P}_{t,ij} = f_t(u_{t,i}, u_{t,j}), u_{t,i} \in [0, 1], u_{t,j} \in [0, 1], \quad 1 \leq i < j \leq n_t.$$

Here $f_s$ and $f_t$ represent the latent graphon function in the source network and target network. Each node $i \in \{1, \ldots, n_s\}$ of the source and target network has a latent position $u_{s,i}$ and $u_{t,i}$, respectively.

Our goal is to estimate the target probability matrix $\mathbf{P}_t$ by leveraging both the observed target graph $\mathbf{A}_t$ and knowledge transferred from the larger source graph $\mathbf{A}_s$, adaptively, when the source is indeed similar to the target.

## 3.2 Proposed Method

**Overview.** Figure 1 shows the workflow of our method GTRANS, which consists of three main steps: (1) Initial graphon estimation: GTRANS begins with observed source and target networks and computes their respective initial estimates $\hat{\mathbf{P}}_s^{ini} \in [0,1]^{n_s \times n_s}$, $\hat{\mathbf{P}}_t^{ini} \in [0,1]^{n_t \times n_t}$. We obtain these initial edge probability matrices using the NS method [65]. These initial estimators capture the basic structure of the respective graphs but may have limited accuracy for the target graph due to its smaller size. (2) Transferring step: We employ optimal transport to calculate the alignment matrix between $\pi$ between the source and target domains. Using this alignment, we transfer the source estimate to the target domain to obtain $\hat{\mathbf{P}}_t^{trans}$, which is further smoothed to obtain $\hat{\mathbf{P}}_t^{trans2}$. If the source-target domain shift (measured by the transport distance $d$) is below a threshold $\delta$, our final estimate is $\hat{\mathbf{P}}^t = \hat{\mathbf{P}}_t^{trans2}$. (3) Debiasing step: When $d > \delta$, indicating a significant domain shift, direct transfer may miss target-specific structures. To correct this, we perform a debiasing step. We compute the residual matrix $\mathbf{R}_t = \hat{\mathbf{P}}_t^{ini} - \hat{\mathbf{P}}_t^{trans2}$, which captures structural patterns present in the target graph but not explained by transfer from the source. However, $\mathbf{R}_t$ is noisy due to the small target sample. To extract a meaningful signal, we apply neighborhood smoothing on $\mathbf{R}_t$ to obtain a denoised residual estimate $\hat{\mathbf{P}}^{res}$. This smoothed correction is then added back to the transferred estimate, yielding the final output $\hat{\mathbf{P}}^t = \hat{\mathbf{P}}_t^{trans2} + \hat{\mathbf{P}}^{res}$.

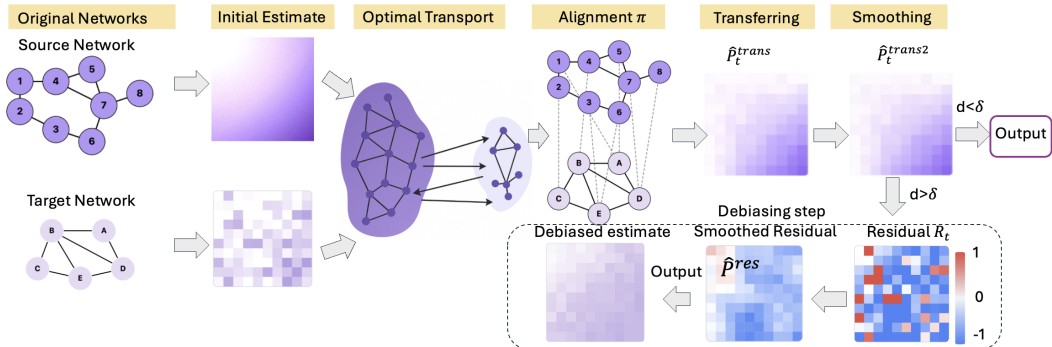

Figure 1: Workflow of GTRANS. Initial estimates from source and target networks are aligned via optimal transport. If source-target distance $d < \delta$, the smoothed transferred estimate is returned. Otherwise, debiasing step is applied to produce the final output.

### 3.2.1 Transferring Step

In this step, we align nodes between the source and target graphs using optimal transport and transfer the source graphon estimator to the target domain. The key idea is to align nodes that exhibit similar connection patterns. We define the oracle alignment as the theoretical correspondence that would be obtained if we had access to the true underlying graph probability matrices. In practice, we don't know true underlying graph probability, researchers have been using observed adjacency matrices for computing alignment estimate. However, each adjacency matrix represents just a single realization of a random graph, which is subject to Bernoulli noise. To address this limitation, we propose using a neighborhood smoothing technique to obtain the initial graphon estimates $\hat{\mathbf{P}}_t^{ini}$ and $\hat{\mathbf{P}}_s^{ini}$, which are shown to be consistent in [65].

In this work, we allow flexibility in choosing between the standard GW distance and its entropic regularized variant EGW for computing the optimal transport plan between two graphs. The choice between GW and EGW depends on computational considerations: For larger source networks with limited computational resources, we recommend using EGW; Otherwise, the standard GW formulation may be preferred for its sharper alignment results. Mathematically, in the GW setting, we compute the optimal transport plan $\hat{\pi} \in [0,1]^{n_s \times n_t}$ by minimizing the following objective function: $\min_{\pi \in \Pi(\mu,\nu)} \sum_{i,j,k,\ell} L(\hat{\mathbf{P}}_s^{ini}(i,k), \hat{\mathbf{P}}_t^{ini}(j,l)) \pi_{ij}\pi_{kl}$. For the EGW variant, we calculate $\hat{\pi}$ similarly but add an entropic regularization term.

**Advantage of using initial graphon estimates for optimal transport.** The advantage of using $\hat{\mathbf{P}}_t^{ini}$ and $\hat{\mathbf{P}}_s^{ini}$ is clearly demonstrated in Figure 2. We illustrate this with a simple example: a 100-node target graph and a 500-node source graph, both generated from the same base graphon function.

We compare the alignment matrices calculated using the original noisy adjacency matrices $\mathbf{A}_t$ and $\mathbf{A}_s$, the initial estimated probability matrices $\hat{\mathbf{P}}_t^{ini}$ and $\hat{\mathbf{P}}_s^{ini}$, and the true underlying probability matrices $\mathbf{P}_t$ and $\mathbf{P}_s$. As shown in Figure 2, the alignment based on raw adjacency matrices is noisy and poorly structured, deviating substantially from the ground-truth alignment. In contrast, the alignment obtained using the initial estimated probability matrices closely approximates the true alignment.

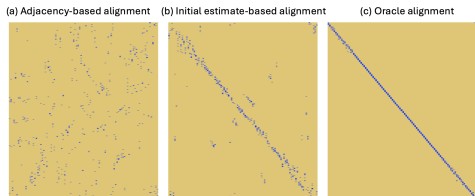

Figure 2: Visualization of alignment matrix.

**Normalization of optimal transport plan.** Each entry $\hat{\pi}_{ij}$ quantifies the strength of the learned correspondence between source node $i$ and target node $j$. Nevertheless, as a transport plan that distributes a total mass of 1 across $n_s \times n_t$ entries, the individual values of $\hat{\pi}_{ij}$ can often be quite small. Thus, we apply column normalization to the transport plan. The resulting matrix, denoted by $\tilde{\pi} \in [0,1]^{n_s \times n_t}$, is scaled such that each column sums to one.

**Projection.** Using the estimated alignment matrix, we transfer the source graphon estimator to the target domain: $\hat{\mathbf{P}}_t^{trans} = \tilde{\pi}^T \hat{\mathbf{P}}_s^{ini} \tilde{\pi}$. This matrix transformation acts as a projection operator, effectively mapping the connectivity patterns encoded in the source probability matrix $\hat{\mathbf{P}}_s^{ini}$ onto the target domain's structure. It produces a weighted aggregation of source graphon values, where the weights reflect the learned node correspondences. To further refine the transfer, we apply neighborhood smoothing to the transferred graphon estimator, obtaining a refined version $\hat{\mathbf{P}}_t^{trans2}$. This additional smoothing step preserves the smoothness properties of the graphon and ensures that the transferred knowledge is not only structurally aligned but also smooth.

### 3.2.2 Debiasing Step

When source and target graphons differ significantly, negative transfer can occur. We quantify this domain difference using the GW between $\hat{\mathbf{P}}_t^{ini}$ and $\hat{\mathbf{P}}_s^{ini}$. When the distance is smaller than the threshold ($d < \delta$), we simply use the smoothed transferred estimator $\hat{\mathbf{P}}_t^{trans2}$ as our final estimator. When this distance exceeds a predetermined threshold ($d > \delta$), we implement the following debiasing procedure to mitigate potential negative transfer effects.

Specifically, we first compute a residual matrix: $\mathbf{R}_t = \hat{\mathbf{P}}_t^{ini} - \hat{\mathbf{P}}_t^{trans2}$. This residual matrix captures (a) target-specific structural patterns not explained by the transferred estimator and (b) random noise due to small sample size of the target graph. To keep the meaningful structure from part (a) but remove the noise from part (b), we then use the neighborhood smoothing method [65]. This method averages information from nearby nodes with similar connection patterns, which smooths out random fluctuations and keeps the stable, informative patterns. The result is a cleaner, smoothed residual estimator, $\hat{\mathbf{P}}_t^{res}$. Finally, we combine the transferred estimator with the smoothed residual to get our final graphon estimator for the target graph: $\hat{\mathbf{P}}_t = \hat{\mathbf{P}}_t^{trans2} + \hat{\mathbf{P}}_t^{res}$. Adding the smoothed residual back allows the model to recover target-specific structural information that was missing from the transferred estimator, without reintroducing the random noise filtered out during smoothing.

Algorithm 1 summarizes our complete procedure, where the optimal transport can be calculated using either GW or its entropic variant EGW, and the choice depends primarily on the computational considerations.

## 4 Theoretical Properties

In this section, we present a stability analysis of the estimated alignment matrix obtained via minimizing the entropic Gromov-Wasserstein distance. Although EGW has been successfully applied in a variety of domains [53, 28], the question of the stability of the optimizer $\pi^*$ with respect to perturbations in the cost or distance matrices $(C, D)$ has received limited attention. Very recently, [66] first established duality theory for EGW, deriving optimal $n^{-1/2}$ empirical convergence rates and proving stability with respect to the regularization parameter and [46] analyzed the a certain aspect of stability of the estimator, when both $C$ and $D$ are euclidean distance matrices based on $m$ and $n$ observations respectively. These analyses do not address the case when (i) $C$ and $D$ are arbitrary cost matrices without the inner-product structure of Euclidean distances, and (ii) the stability of the

EGW estimator under perturbations of the cost functions, both of which are particularly relevant to our transfer learning framework. Recall that in the GTRANS algorithm, the estimated alignment matrix $\hat{\pi}$ is obtained using EGW with inputs $C = \hat{\mathbf{P}}_s^{\text{ini}}$ and $D = \hat{\mathbf{P}}_s^{\text{ini}}$. As a result, these matrices cannot be interpreted as Euclidean distance matrices between point clouds, since they do not arise from pairwise distances in an inner-product space. It is therefore natural to ask how far $\hat{\pi}$ is from the *oracle* alignment matrix $\pi^*$, which minimizes the same objective using the true connectivity matrices $C = \mathbf{P}_s$ and $D = \mathbf{P}_t$. While the stability of convex optimization problems has been extensively studied (see, e.g., [6] for an overview of techniques and results), these tools do not directly apply to our setting due to the nonconvex nature of the EGW objective. Our subsequent theorem showcases that the distance between $\hat{\pi}$ and $\pi^*$ can be upper bounded by a distance between $\hat{\mathbf{P}}_s^{\text{ini}}$ and $\mathbf{P}_s$, and $\hat{\mathbf{P}}_t^{\text{ini}}$ and $\mathbf{P}_t$:

**Theorem 4.1.** *Let $\hat{\pi}$ and $\pi^*$ is the solution of EGW optimization problem with $(C, D) = (\hat{\mathbf{P}}_s^{\text{ini}}, \hat{\mathbf{P}}_t^{\text{ini}})$ and $(C, D) = (\mathbf{P}_s, \mathbf{P}_t)$ respectively. If the penalty parameter $\epsilon$ satisfies: $\|\pi^*\|_\infty \leq \frac{\epsilon}{C_1 \|\mathbf{P}_s \otimes \mathbf{P}_t\|_{\text{op}}}$, for some $C_1 > 2$, then,*

$$\|\hat{\pi} - \pi^*\|_F \leq C_2 \frac{\left\| (\mathbf{P}_s \otimes \mathbf{P}_t) - (\hat{\mathbf{P}}_s^{ini} \otimes \hat{\mathbf{P}}_t^{ini}) \right\|_{\text{op}}}{\|\mathbf{P}_s \otimes \mathbf{P}_t\|_{\text{op}}}.$$

*as soon as $\|\mathbf{P}_s - \hat{\mathbf{P}}_s^{ini}\|_\infty + \|\mathbf{P}_t - \hat{\mathbf{P}}_t^{ini}\|_\infty$ is less than a certain cutoff (mentioned explicitly in the proof), for some universal constant $C_2$ that depends on $C_1$.*

We would like to highlight that our analysis is quite general as it does not rely on any structure of $(\mathbf{P}_s, \mathbf{P}_t)$, which makes our analysis different from [66, 46]. The lower bound on the $\epsilon$ is needed to bring some local convexity structure of the problem near the oracle optimizer $\pi^*$. Such a lower bound condition for $\epsilon$ is typically adopted in literature, e.g., see [49].

The following result shows that the empirical GW distance between initial estimators $\hat{\mathbf{P}}_S^{\text{init}}$ and $\hat{\mathbf{P}}_T^{\text{init}}$ closely approximates the population GW distance between $\mathbf{P}_S$ and $\mathbf{P}_T$, up to small estimation error.

**Theorem 4.2.** *Let $\delta_{n_S}$ and $\delta_{n,T}$ denote the estimation error of $\mathbb{P}_S$ and $\mathbb{P}_T$ respectively with respect to average squared Frobenious norm, i.e.*

$$\mathbb{P}\left( \frac{1}{n_S^2}\|\hat{\mathbf{P}}_S^{\text{init}} - \mathbf{P}_S\|_F^2 \leq \delta_{n_S} \right) \geq 1 - \epsilon_n, \quad \mathbb{P}\left( \frac{1}{n_T^2}\|\hat{\mathbf{P}}_T^{\text{init}} - \mathbf{P}_T\|_F^2 \leq \delta_{n_T} \right) \geq 1 - \epsilon_n.$$

*Define $\delta_n = \delta_{n_S} + \delta_{n_T}$. Then, on the intersection of the above events, we have:*

$$\text{GW}_2^2(\mathbf{P}_s, \mathbf{P}_t) \leq 4 \left( \text{GW}_2^2(\hat{\mathbf{P}}_S^{\text{init}}, \hat{\mathbf{P}}_T^{\text{init}}) + \delta_n \right)$$

$$\text{GW}_2^2(\hat{\mathbf{P}}_S^{\text{init}}, \hat{\mathbf{P}}_T^{\text{init}}) \leq 4 \left( \text{GW}_2^2(\mathbf{P}_s, \mathbf{P}_t) + \delta_n \right).$$

We next justify the theoretical validity of the alignment step. The following result shows that applying the optimal GW coupling preserves $L_2$ proximity between the source and target graphons.

**Theorem 4.3.** *Let $\pi^*$ be the optimal coupling of $(f, g)$ for GW-distance, i.e.*

$$\pi^* = \arg\min_{\pi \in \Gamma} \int \int (f(x, x') - g(y, y'))^2 \, d\pi(x, y) \, d\pi(x', y').$$

*Define $\tilde{g}$ (population analogue of $\hat{\pi}^\top \hat{P}_s \hat{\pi}$) as:*

$$\tilde{g}(y, y') = \int_{[0,1]^2} \pi^*(y, x) f(x, x') \pi^*(x', y) \, dx \, dx'.$$

*Then we have,*

$$\|\tilde{g} - g\|_2^2 \leq GW_2^2(f, g).$$

*As a consequence, when $\text{GW}_2(f, g)$ is small, then $\tilde{g}$ is close to g, which validates the alignment strategy.*

**Remark 4.4.** *The estimation error $\delta_{n_S}, \delta_{n_T}$ depends on the estimation strategy and the underlying data-generating process. For example, of $\mathbf{P}_s$ and $\mathbf{P}_T$ are generated from a $\alpha$-Hölder graphon, then it is possible to construct $\hat{\mathbf{P}}_s$ and $\hat{\mathbf{P}}_T$ such that $\delta_n \asymp n^{-2\alpha/(\alpha+1)}$ for $0 < \alpha < 1$ and $\delta_n \asymp (\log n)/n$ for $\alpha \geq 1$ (e.g., see [19]) for $n \in \{n_S, n_T\}$, albeit they are NP-hard to compute. Later, [65] proposed a neighborhood smoothing based approach to obtain $(\hat{\mathbf{P}}_S, \hat{\mathbf{P}}_t)$ with $\delta_n \asymp \sqrt{\log n/n}$ for $n \in \{n_S, n_T\}$.*

# 5 Experiments

## 5.1 Simulation Experiments

**Simulation setup.** To evaluate the effectiveness of our proposed method, GTRANS, we conduct simulation studies using ten representative graphon functions. Figure 3 illustrates the probability matrices $\mathbf{P}$ of 500 nodes generated for Graphons 1 to 10, where the rows and the columns are ordered by latent position $u$. Graphons 1-5 are graphon functions used in SAS [9], and graphons 6-10 are used in graphon functions used in ICE [45]. The details about the structure of these graphon functions can be found in Section G.1.

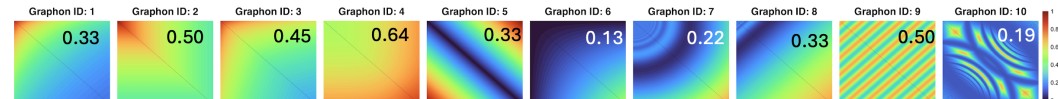

Figure 3: Heatmap of true probability matrix for graphon 1-10 (from left to right), where high values of probability are colored in red, and low values of probability are colored in blue. The number in each heatmap means the average connecting probability.

**Baseline.** To evaluate the effectiveness of our proposed method GTRANS, we compare its performance with four other existing graphon estimation methods which use target data only: neighborhood smoothing (NS) [65], sorting and smoothing (SAS) [9], universal singular value thresholding algorithm (USVT) [11], and iterative connecting probability estimation (ICE) [45]. We report results for our method implemented with both standard and entropic Gromov-Wasserstein alignment, denoted as GTRANS-GW and GTRANS-EGW, respectively. For all experiments, we evaluate performance using the MSE. We performed 50 independent simulation runs to calculate the averaged MSE.

In this simulation, we aim to answer three key questions: (1) Impact of source sample size: How does the performance of GTRANS improve as the sample size in the source domain increases? (2) Impact of domain shift: How robust is GTRANS against negative transferring when transferring between two different graphons? (3) Impact of density shift: How effective is GTRANS when transferring from a dense network to a sparse network or vice versa?

**Asymptotic performance.** To investigate the impact of source sample sizes, we vary the sample size of source data $n_s \in \{100, 200, \ldots, 1000\}$ while fixing other parameters: size of the target data $n_t = 50$. We introduce a small perturbation between source and target graphons: $f_t(u, v) = f_s(u, v) + \xi$ where $\xi$ generated uniformly from $U(-0.01, 0.01)$. Figure 4 displays MSE results across ten different graphon types. Several key observations: (1) Both GTRANS-GW and GTRANS-EGW consistently outperforms all baseline methods across all graphon types, demonstrating the effectiveness of our transfer learning approach. (2) We observe a clear decreasing trend in MSE as source sample size increases for both of GTRANS-GW and GTRANS-EGW, while other target only methods (including NS, USVT, ICE, SAS) show flat pattern because they can not borrow strength from source sample. (3) GTRANS-GW and GTRANS-EGW achieve comparable performance overall, though in some cases, the entropic regularization in GTRANS-EGW yields lower MSE, benefiting from entropic regularization to enhance alignment stability. Figure 5 further illustrates our method's effectiveness (using GW for demonstration) when transferring from a 500-node source to a 50-node target network (both from graphon 8). Panel (a) displays the noisy target-only estimation. Panel (b) shows the accurate source estimation (left), significantly improved transferred result (middle), and final estimation (right). Since the source and target domains are identical, the optimal transport distance is small, and the final estimate equals the transferred result without requiring debiasing.

**Transfer between different graphons.** To assess the robustness of GTrans to domain differences, we fix $n_s = 500$ and $n_t = 50$, and consider transfer learning between different graphon pairs. We consider transfer between similar graphon pairs (such as 1 and 3, or 6 and 7, 7 and 8), as well as dissimilar graphon pairs (such as 8 and 9, or 5 and 10). Table 1 presents the MSE results for these transfer scenarios compared to target-only methods. We observe several key patterns in the results: (1) For similar graphon pairs, both GTRANS-GW and GTRANS-EGW achieves substantial improvement over target-only methods. This demonstrates that when graphons share structural characteristics, knowledge transfer remains highly effective even between different graphon functions. Figure 5(c) provides visual evidence of effective knowledge transfer in the (from graphon 7 to 8) scenario. Both graphons share smooth gradient-based transition patterns. The middle panel in (c) shows that the

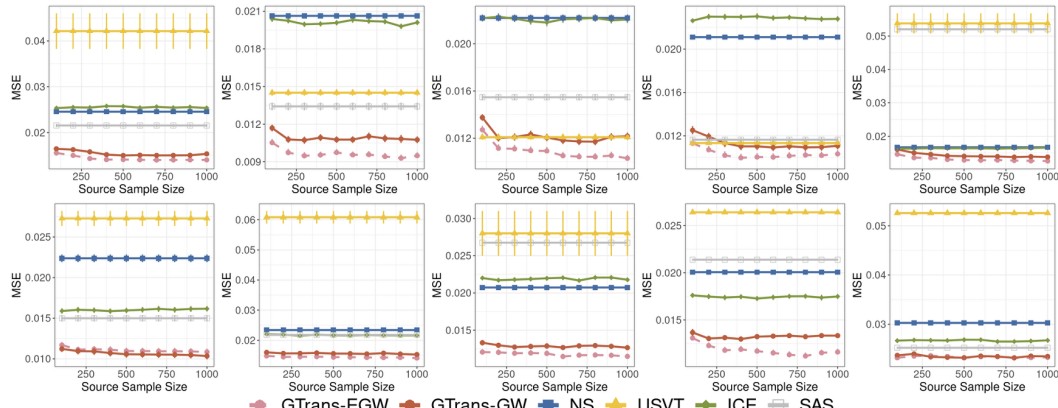

Figure 4: MSE performance of five methods as source network size increases from 100-1000 nodes, with error bars representing ±0.1 standard deviations. GTRANS-GW (red circles with solid line), GTRANS-EGW (pink circles with dashed line), NS (blue squares with solid line), USVT (yellow triangles with solid line), ICE (green diamonds with solid line), SAS (gray hollow squares with solid line). Top row: graphons 1-5; bottom row: graphons 6-10.

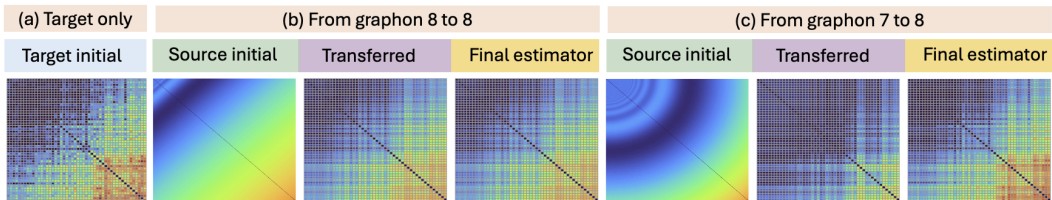

Figure 5: Visual comparison of GTRANS performance with different source graphons when estimating target graphon 8. (a) target-only initial estimator using only limited target data; (b-c) three panels showing source initial estimator, transferred estimator, and final estimator from different source graphons ($8 \rightarrow 8, 7 \rightarrow 8$ respectively).

transferred estimator successfully preserves the source's smooth structural regularity while adapting to the target's diagonal gradient, evident from the emergence of a grid-like texture where intensity gradually decreases diagonally. The final estimator in the right panel of (c) exhibits both the structural smoothness inherited from the source and a more precise alignment with the target graphon's latent geometry, highlighting the advantage of debiasing step. (2) For transfers between graphon pairs with larger difference, GTRANS maintains similar performance compared to its target-only counterpart NS method. This robustness is due to our adaptive debiasing mechanism, which effectively mitigates negative transfer while preserving beneficial knowledge.

**Impact of density shift.** To investigate the performance of GTRANS to density differences between source and target domains, we examine scenarios with varying levels of perturbation between source and target graphons: $f_s(u,v) = f_t(u,v) + \xi$ where $\xi \sim U(0,\lambda)$ if $\lambda > 0$, $\xi \sim U(\lambda,0)$ if $\lambda < 0$. We vary $\lambda$ from $\{-0.5, -0.45, \ldots, 0.5\}$, and truncate the probability if it is larger than 1 or smaller than 0. This allows us to simulate scenarios where the target network is either sparser ($\lambda > 0$) or denser

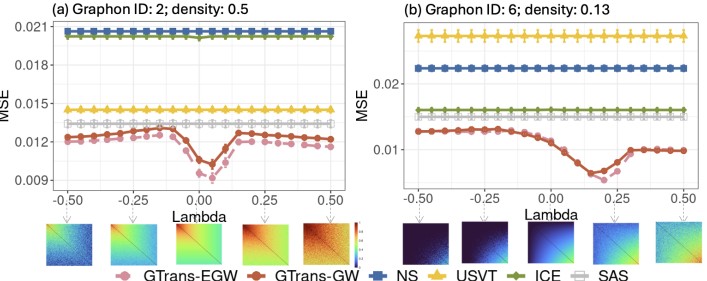

Figure 6: MSE comparison of GTRANS-GW, GTRANS-EGW, NS, ICE, USVT, and SAS under density shifts $\lambda$ for Graphon 2 (0.5) and Graphon 6 (0.13), with error bars representing ±0.1 standard deviations. Bottom panels show adjacency matrix transformations at $\lambda = -0.50, -0.25, 0.00, 0.25, 0.50$.

($\lambda < 0$) than the source. Figure 6 presents MSE results for two representative graphons (ID 2 and ID 6). Additional results can be found in Figure E1. For the dense Graphon 2 (density = 0.5), GTRANS-GW and GTRANS-EGW both show a symmetric U-shaped pattern centered almost around zero, indicating that performance degrades equally whether transferring from a denser or sparser source. In contrast, Graphon 6 is a sparse network (density = 0.13). Here, the performance curve of GTRANS-GW and GTRANS-EGW achieve its lowest error at a positive value of $\lambda$. This indicates that when the target graph is dense, the impact of density shift on transfer learning is symmetric around zero. However, when the target graph is sparse, transfer performance becomes more sensitive to the direction of the density shift, and transferring from a slightly denser source network yields better results. The underlying reason might be: (1) a denser source network typically exhibits a higher signal-to-noise ratio, yielding a higher-quality initial graphon estimator.

(2) When the source network is sparse, $\hat{\mathbf{P}}_s^{ini}$ will also be sparse—potentially to an extreme degree where it might even be (approximately) a zero matrix. This severely limits the information available for transfer, regardless of how well the transport plan aligns the domains.

Table 1: MSE comparison for transfer learning between different graphons ($n_s = 500$, $n_t = 50$)

| Similarity | Scenario | GTRANS-GW | GTRANS-EGW | NS | USVT | ICE | SAS |
|---|---|---|---|---|---|---|---|
| | $7 \rightarrow 6$ | **0.9 $\pm$ 0.2** | **1.0 $\pm$ 0.3** | 2.1 $\pm$ 0.3 | 3.0 $\pm$ 0.9 | 1.7 $\pm$ 0.3 | 1.6 $\pm$ 0.4 |
| | $6 \rightarrow 7$ | **1.8 $\pm$ 0.3** | **1.8 $\pm$ 0.3** | 2.3 $\pm$ 0.3 | 5.9 $\pm$ 2.7 | 2.2 $\pm$ 0.3 | 2.4 $\pm$ 0.6 |
| Similar | $8 \rightarrow 7$ | **1.3 $\pm$ 0.3** | **1.4 $\pm$ 0.3** | 2.3 $\pm$ 0.3 | 5.9 $\pm$ 2.7 | 2.2 $\pm$ 0.3 | 2.4 $\pm$ 0.6 |
| | $7 \rightarrow 8$ | 1.6 $\pm$ 0.2 | 1.6 $\pm$ 0.2 | 2.0 $\pm$ 0.2 | **1.5 $\pm$ 0.2** | 2.1 $\pm$ 0.2 | 2.8 $\pm$ 0.7 |
| | $3 \rightarrow 1$ | **1.6 $\pm$ 0.5** | **1.7 $\pm$ 0.2** | 2.5 $\pm$ 0.3 | 5.1 $\pm$ 4.6 | 2.7 $\pm$ 0.3 | 2.1 $\pm$ 0.3 |
| | $1 \rightarrow 3$ | 1.6 $\pm$ 0.3 | 1.5 $\pm$ 0.3 | 2.3 $\pm$ 0.3 | **1.2 $\pm$ 0.2** | 2.2 $\pm$ 0.3 | 1.6 $\pm$ 0.3 |
| | $8 \rightarrow 9$ | 1.9 $\pm$ 0.4 | 1.9 $\pm$ 0.3 | 2.0 $\pm$ 0.2 | 2.6 $\pm$ 0.1 | **1.7 $\pm$ 0.2** | 2.1 $\pm$ 0.1 |
| Different | $9 \rightarrow 8$ | 2.0 $\pm$ 0.2 | 1.9 $\pm$ 0.2 | 2.0 $\pm$ 0.2 | **1.5 $\pm$ 0.2** | 2.1 $\pm$ 0.2 | 2.8 $\pm$ 0.7 |
| | $10 \rightarrow 5$ | **1.6 $\pm$ 0.2** | **1.5 $\pm$ 0.2** | 1.7 $\pm$ 0.1 | 3.5 $\pm$ 1.5 | 1.7 $\pm$ 0.2 | 5.5 $\pm$ 1.0 |
| | $5 \rightarrow 10$ | **2.6 $\pm$ 0.3** | **2.5 $\pm$ 0.3** | 3.1 $\pm$ 0.6 | 5.3 $\pm$ 0.6 | 2.8 $\pm$ 0.3 | 2.6 $\pm$ 0.3 |

**Selection of Hyperparameters.** GTRANS-GW involves one hyperparameter: threshold $\delta$ while GTRANS-GW involves two hyperparameters: threshold $\delta$ and regularization parameters $\epsilon$. In this paper, we employed a network cross-validation procedure developed by [33] for hyperparameter tuning. (1) Selection of $\delta$: Specifically, when implementing GTRANS with either GW or EGW, we choose $\delta$ from a candidate set $\{0.1, 0.11, 0.12, ....., 0.5\}$, and use the cross-validation procedure to select the value that minimizes held-out prediction error. Figures E3a and E3b illustrate the optimal thresholds identified across different simulation scenarios for GW and EGW, respectively. As we can see: most GW scenarios work best when $\delta = 0.15$, while most EGW scenarios work best when $\delta = 0.18$. While cross-validation is preferred, we recommend default values of $\delta = 0.15$ for GW and $\delta = 0.18$ for EGW when tuning is computationally infeasible in practice. (2) Selection of $\epsilon$: When implementing EGW, we select the optimal regularization parameter from candidates $\{0.001, 0.005, 0.01, \ldots, 0.1\}$. Figures E4 illustrates the MSE comparison across different regularization parameters $\epsilon$, highlighting that $\epsilon = 0.01$ is consistently effective for most graphons.

## 5.2 Application to Real Data

Evaluating probability matrix estimation methods on real networks directly is difficult, since the true probability matrix is unknown. We assess the practical utility of our method by applying it to two downstream applications: graph classification and link prediction. Due to page limitations, we only show results for graph classification. Link prediction results are provided in Appendix F.2.

Graph classification is fundamental to network analysis, but often suffers from limited labeled data. [22] proposed G-Mixup, which augments datasets by interpolating between graphons of different classes to generate synthetic training graphs. The quality of this augmentation critically depends on the accuracy of the underlying graphon estimation. When the network size is small, conventional graphon estimators often produce poor results. For example, IMDB-B and IMDB-M datasets used in [22] have only 19.77 and 13.00 average nodes per graph, respectively.

**Datasets.** To address this challenge, we implemented GTRANS to enhance G-Mixup for graph classification by transferring knowledge from larger networks. In our experiments, we consider two co-actor graph datasets as targets: two-class IMDB-BINARY and three-class IMDB-MULTI, both characterized by small graph sizes. As candidate sources, we consider: (1) three-class COLLAB

(average 74.49 nodes per graph), a collaboration network derived from scientific authorship data [39], and (2) two-class Reddit-Binary (average 429 nodes per graph), comprising Reddit user interaction threads [39]. Additionally, we examine a bioinformatics setting by transferring from two-class D&D (average 284.32 nodes) to two-class PROTEINS-Full (average 25.22 nodes) [39], both consisting of protein structure graphs.

**Results.** We adopt the same Graph Convolutional Network (GCN) architecture as used in [22], using the same hyperparameters and training procedures for all benchmark comparisons. Full implementation details are provided in Appendix F.1. For each target dataset, we split the dataset into train/validation/test data by $70\%/10\%/20\%$. We report the test accuracy on ten runs. Tables 2 report test accuracies for our proposed GTRANS method against baselines, including NS [65], USVT [11], SAS [9], ICE [45], GWB [57], IGNR [55], SIGL [4], and graphlet [44, 48]. For GWB/IGNR/SIGL we follow the original implementations and keep hyperparameters unchanged. The graphlet baseline is implemented via GraKeL's Graphlet Kernel with an SVM on the precomputed kernel matrix; we use graphlet sampling with 700 subgraphs per graph for scalability. On IMDB-Binary GTRANS-GW achieves 76.30% (transfer from Reddit-Binary) and 76.25% (transfer from COLLAB), while GTRANS-EGW further improves to 76.80% and 77.50%, respectively, outperforming all baselines by **2–5%**. On IMDB-Multi, GTRANS-GW with COLLAB reaches 50.47%, and GTRANS-EGW achieves 51.27% with Reddit-Binary as the source, demonstrating clear superiority over all other methods. For PROTEINS-Full, transferring from D&D yields 69.33% with GTRANS-GW and 68.52% with GTRANS-EGW, consistently outperforming baseline methods by 3–6 points. These results confirm that both GTRANS-GW and GTRANS-EGW effectively boost classification accuracy by leveraging structural knowledge from larger networks.

Table 2: Graph classification accuracy (%) across three target datasets (mean $\pm$ std) compared against graphon estimation baselines.

| Source | Target | GTRANS-GW | GTRANS-EGW | NS | USVT | SAS |
|--------|--------|-----------|------------|-----|------|-----|
| Reddit-B | IMDB-B | $76.30 \pm 2.35$ | $\mathbf{76.80 \pm 1.52}$ | $72.90 \pm 2.10$ | $73.85 \pm 2.40$ | $74.25 \pm 1.93$ |
| COLLAB | IMDB-B | $76.25 \pm 2.06$ | $\mathbf{77.50 \pm 2.13}$ | $72.90 \pm 2.10$ | $73.85 \pm 2.40$ | $74.25 \pm 1.93$ |
| Reddit-B | IMDB-M | $49.10 \pm 1.33$ | $\mathbf{51.27 \pm 1.98}$ | $43.80 \pm 2.59$ | $48.00 \pm 2.93$ | $44.10 \pm 2.05$ |
| COLLAB | IMDB-M | $\mathbf{50.47 \pm 1.42}$ | $50.23 \pm 0.92$ | $43.80 \pm 2.59$ | $48.00 \pm 2.93$ | $44.10 \pm 2.05$ |
| D&D | PROTEINS | $69.33 \pm 2.55$ | $68.52 \pm 1.59$ | $63.18 \pm 1.94$ | $65.11 \pm 2.21$ | $65.25 \pm 1.85$ |

| Source | Target | ICE | GWB | IGNR | SIGL | Graphlet |
|--------|--------|-----|-----|------|------|----------|
| Reddit-B | IMDB-B | $74.30 \pm 2.16$ | $75.30 \pm 2.19$ | $74.35 \pm 3.51$ | $73.50 \pm 2.62$ | $61.10 \pm 2.23$ |
| COLLAB | IMDB-B | $74.30 \pm 2.16$ | $75.30 \pm 2.19$ | $74.35 \pm 3.51$ | $73.50 \pm 2.62$ | $61.10 \pm 2.23$ |
| Reddit-B | IMDB-M | $43.90 \pm 1.27$ | $47.70 \pm 2.53$ | $47.50 \pm 3.32$ | $49.13 \pm 3.04$ | $39.37 \pm 1.62$ |
| COLLAB | IMDB-M | $43.90 \pm 1.27$ | $47.70 \pm 2.53$ | $47.50 \pm 3.32$ | $49.13 \pm 3.04$ | $39.37 \pm 1.62$ |
| D&D | PROTEINS | $65.38 \pm 1.84$ | $63.45 \pm 2.96$ | $65.87 \pm 2.56$ | $67.13 \pm 2.34$ | $\mathbf{70.11 \pm 2.12}$ |

# 6 Conclusion and Future Work

In this paper, we proposed GTRANS, a novel transfer learning framework to estimate the edge connecting probability, addressing the challenge of limited data availability in the target network. By leveraging Gromov-Wasserstein optimal transport, our method aligns latent node structures across source and target domains and effectively transfers graphon information to improve estimation accuracy. Extensive experiments on both synthetic and real-world networks demonstrate that our approach consistently outperforms existing baselines, particularly in scenarios involving small, sparse graphs. By improving estimation accuracy, our framework facilitates various downstream tasks, including data augmentation and link prediction. Despite these promising results, several limitations remain. First, our current framework assumes a single source graph. Extending it to incorporate multiple source networks could further enhance robustness by leveraging diverse structural priors. Second, GTRANS is designed for static graphs; future extensions to dynamic or multi-layer networks would enable modeling of time-evolving or multi-modal dependencies. Third, our current formulation does not incorporate node-level covariates. Integrating such covariates could provide valuable auxiliary information for alignment, particularly in domains where topological structure alone may be insufficient for effective transfer. These directions can further enhance the flexibility and generalizability of transfer-based graphon estimation.

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

# Appendix for "Transfer Learning on Edge Connecting Probability Estimation Under Graphon Model"

**Yuyao Wang**
Boston University
yuyaow@bu.edu

**Yu-Hung Cheng**
Boston University
yhcheng@bu.edu

**Debarghya Mukherjee**
Boston University
mdeb@bu.edu

**Huimin Cheng**
Boston University
huimin23@bu.edu

## A   Notation Table

Table A1: Notations used throughout the paper

| Notation | Description |
|---|---|
| $n_s, n_t$ | Number of nodes in the source and target graphs |
| $\mathbf{A}_s, \mathbf{A}_t$ | Adjacency matrices of source and target graphs, $\mathbf{A} \in \{0, 1\}^{n \times n}$ |
| $\mathbf{P}_s, \mathbf{P}_t$ | True connection probability matrices for source and target graphs |
| $\hat{\mathbf{P}}_s^{\text{ini}}, \hat{\mathbf{P}}_t^{\text{ini}}$ | Initial estimators via neighborhood smoothing |
| $\hat{\mathbf{P}}_t^{\text{trans}}, \hat{\mathbf{P}}_t^{\text{trans2}}$ | Transferred estimator before and after smoothing |
| $\hat{\mathbf{P}}_t^{\text{res}}$ | Smoothed residual graphon estimator |
| $\hat{\mathbf{P}}_t$ | Final graphon estimator for the target domain |
| $\hat{\mathbf{P}}_{ij}$ | Estimated connecting probability between node $i$ and $j$ |
| $\text{Ber}(p)$ | Bernoulli distribution with parameter $p$ |
| $f_s, f_t$ | Latent graphon functions for source and target domains |
| $u_{s,i}, u_{t,i}$ | Latent position of node $i$ in $[0, 1]$ |
| $\Pi(\mu, \nu)$ | Set of couplings (transport plans) between $\mu$ and $\nu$ |
| $\hat{\pi} \in [0, 1]^{n_s \times n_t}$ | The optimal transport plan estimated by Gromov-Wasserstein |
| $\tilde{\pi} \in [0, 1]^{n_s \times n_t}$ | Normalized transport plan |
| $\delta$ | Threshold for triggering the debiasing step |
| $\epsilon$ | Entropic regularization parameter in Gromov-Wasserstein optimization |
| $\lambda$ | Density shift parameter for source-target perturbation. |
| $\text{KL}(\pi \mid \mu \otimes \nu)$ | Kullback-Leibler divergence: $\text{KL}(\pi \mid \mu \otimes \nu) = \sum_{i,j} \pi_{ij} \log \frac{\pi_{ij}}{\mu_i \nu_j}$ |
| $\mathbf{P}_s \otimes \mathbf{P}_t$ | Kronecker product of $\mathbf{P}_s$ and $\mathbf{P}_t$ |
| $\|\cdot\|_F$ | Frobenius norm: $\|\mathbf{X}\|_F = \sqrt{\sum_{i,j} X_{ij}^2}$ |
| $\|\cdot\|_2$ | $L_2$ norm: $\|f - g\|_2 = \left( \int_0^1 \int_0^1 (f(u,v) - g(u,v))^2 \, du \, dv \right)^{1/2}$ |
| $\|\cdot\|_\infty$ | Infinity norm: $\|\mathbf{X}\|_\infty = \max_{i,j} |X_{ij}|$ |
| $\|\cdot\|_{\text{op}}$ | Operator norm: $\|\mathbf{X}\|_{\text{op}} = \sigma_{\max}(\mathbf{X})$ |

39th Conference on Neural Information Processing Systems (NeurIPS 2025).

# B Proof of Theorem 4.1

In this proof, we will use the notation $B(x;\tau)$ to denote a ball of radius $\tau$ around $x$, with respect to some appropriate distance, which will be clear from the context. Before delving into the proof, let us first recall the definition of strong convexity:

**Definition B.1.** *A function $f$ is said to be strongly convex in a neighborhood around $x_*$ (namely $B(x_*;\tau)$), if $f$ satisfies:*

$$f(y) \geq f(x) + \langle y - x, \nabla f(x) \rangle + \frac{\mu}{2}\|y - x\|_2^2$$

Our proof relies on an application of the following lemma:

**Lemma B.2.** *Suppose $f$ is a convex function which is minimized at $x_*$ and furthermore it is $\mu$-strongly convex on $B(x_*;\tau) = \{x : \|x - x_*\|_2 \leq \tau\}$. Consider a perturbation $g$ of $f$ such that i) $\|f - g\|_\infty \leq \delta$ and ii) $g - f$ is Lipschitz with Lipschitz-constant $\kappa$. If $\delta \leq \mu\tau^2/4$, we have:*

$$\|x_* - y_*\|_2 \leq \frac{2\kappa}{\mu} \equiv \frac{2\|f - g\|_{\text{Lip}}}{\mu}.$$

*Proof.* We divide the proof of Lemma B.2 into two smaller lemmas. The first lemma is as follows:

**Lemma B.3.** *Consider two functions $f$ and $g$, such that $f - g$ is $\kappa$-Lipschitz. Suppose $x_*$ and $y_*$ are minimizer of $f$ and $g$ respectively. If $f$ is strongly convex on $B(x_*;\tau)$, and $y_* \in B(x_*;\tau)$, then*

$$\|x_* - y_*\|_2 \leq \frac{2\kappa}{\mu}.$$

*Proof.* The proof essentially follows from Proposition 4.32 of [6]. Here, we sketch the proof for the ease of the readers. As $y_*$ is the minimizer of $g$, we have:

$$f(y_*) - f(x_*) = (f - g)(y_*) - (f - g)(x_*) + \underbrace{g(y_*) - g(x_*)}_{\leq 0}$$

$$\leq (f - g)(y_*) - (f - g)(x_*) \leq \kappa\|x_* - y_*\|_2.$$

On the other, from the strong convexity of $f$ on $B(x_*;\tau)$ and the assumption that $y_* \in B(x_*;\tau)$, we have:

$$f(y_*) \geq f(x_*) + \frac{\mu}{2}\|x_* - y_*\|_2^2,$$

where we use the fact that $\nabla f(x_*) = 0$. Combining these two equations, we conclude:

$$\frac{\mu}{2}\|x_* - y_*\|_2^2 \leq f(y_*) - f(x_*) \leq \kappa\|x_* - y_*\|_2 \implies \|x_* - y_*\|_2 \leq \frac{2\kappa}{\mu}.$$

This completes the proof. $\qquad\square$

One of the requirements of Lemma B.3 is that $y_* \in B(x_*,\tau)$. The following lemma gives a sufficient condition for this condition to be satisfied:

**Lemma B.4.** *Assume $f$ is $\mu$-strongly convex on $B(x_*;\tau)$. If $\|f - g\|_\infty \leq \delta$, $\delta \leq \tau\mu^2/4$, then the minimizer of $g$ also lies in $B(x_*;\tau)$.*

*Proof.* Suppose $y_* \notin B(x_*;\tau)$. Then $\|x_* - y_*\| > \tau$. Therefore, $\exists t \in (0,1)$ such that $x(t) = tx_* + (1 - t)y_*$ lies on the boundary, i.e., $\|x_* - x(t)\|_2 = \tau$. By the strong convexity of $f$ on $B(x_*;\tau)$, we have:

$$f(x(t)) \geq f(x_*) + \frac{\mu\tau^2}{2}.$$

On the other hand:

$$f(x(t)) \leq g(x(t)) + \delta \leq tg(x_*) + (1 - t)g(y_*) + \delta \leq g(x_*) + \delta.$$

Hence, we can conclude that:

$$g(x_*) \geq f(x_*) + \frac{\mu\tau^2}{2} - \delta$$

This immediate contradicts the fact that $|g(x_*) - f(x_*)| \leq \delta$ as $\tau^2 > 4\delta/\mu$. This completes the proof. $\qquad\square$

Finally, the claim in Lemma B.2 is established by combining the arguments of B.3 and Lemma B.4. $\qquad\square$

We use Lemma B.2 to complete the proof of Theorem 4.1. With $\mu = (1/n_s, \cdots, 1/n_s)$ and $\nu = (1/n_t, \cdots, 1/n_t)$ The oracla EGW optimization problem (with respect to $\mathbf{P}_s, \mathbf{P}_t$) can be written as:

$$\mathcal{L}(\pi)$$

$$= \frac{1}{2}\sum_{ijkl}(\mathbf{P}_{s,ik} - \mathbf{P}_{t,jl})^2\pi_{ij}\pi_{kl} + \epsilon\sum_{ij}\pi_{ij}\log\left(n_s n_t\,\pi_{ij}\right)$$

$$= \frac{1}{2}\sum_{ijkl}\mathbf{P}_{s,ik}^2\pi_{ij}\pi_{kl} + \frac{1}{2}\sum_{ijkl}\mathbf{P}_{t,jl}^2\pi_{ij}\pi_{kl} - \sum_{ijkl}\mathbf{P}_{s,ik}\mathbf{P}_{t,jl})\pi_{ij}\pi_{kl} + \epsilon\sum_{ij}\pi_{ij}\log\pi_{ij} + \log\left(n_s n_t\right)\epsilon\underbrace{\sum_{ij}\pi_{ij}}_{=1}$$

$$= \frac{1}{2}\|\mathbf{P}_s\|_F^2 + \frac{1}{2}\|\mathbf{P}_t\|_F^2 - \sum_{ijkl}\mathbf{P}_{s,ik}\mathbf{P}_{t,jl}\pi_{ij}\pi_{kl} + \epsilon\sum_{ij}\pi_{ij}\log\pi_{ij} + \epsilon\log\left(n_s n_t\right).$$

It is immediate that the first, second, and fourth terms do not contribute to the estimation of $\pi$. Ignoring them, we redefine the oracle loss function as:

$$f(\pi) = -\sum_{ijkl}\mathbf{P}_{s,ik}\mathbf{P}_{t,jl}\pi_{ij}\pi_{kl} + \epsilon\sum_{ij}\pi_{ij}\log\pi_{ij} = -2\mathsf{tr}(\pi^\top\mathbf{P}_s\pi\mathbf{P}_t) + \epsilon\sum_{ij}\pi_{ij}\log\pi_{ij}$$

Similary, we define $g(\cdot)$ as the sample version (with respect to $(\hat{\mathbf{P}}_s^{\mathrm{init}}, \hat{\mathbf{P}}_t^{\mathrm{init}})$):

$$g(\pi) = -\sum_{ijkl}\hat{\mathbf{P}}_{s,ik}\hat{\mathbf{P}}_{t,jl}\pi_{ij}\pi_{kl} + \epsilon\sum_{ij}\pi_{ij}\log\pi_{ij} = -2\mathsf{tr}(\pi^\top\hat{\mathbf{P}}_s\pi\hat{\mathbf{P}}_t) + \epsilon\sum_{ij}\pi_{ij}\log\pi_{ij}\,.$$

For notation simplicity define $\delta_1 = \|\mathbf{P}_s - \hat{\mathbf{P}}_s\|_{\infty,\infty}$ and $\delta_2 = \|\mathbf{P}_t - \hat{\mathbf{P}}_t\|_{\infty,\infty}$. We have:

$$\frac{1}{2}\left|f(\pi) - g(\pi)\right| = \left|\mathsf{tr}(\pi^\top\mathbf{P}_s\pi\mathbf{P}_t^\top) - \mathsf{tr}(\pi^\top\hat{\mathbf{P}}_s\pi\hat{\mathbf{P}}_t^\top)\right|$$

$$= \left|\sum_{i,j}\pi_{ij}\left(\mathbf{P}_s\pi\mathbf{P}_t^\top - \hat{\mathbf{P}}_s\pi\hat{\mathbf{P}}_t^\top\right)_{ij}\right|$$

$$\leq \max_{i,j}\left|\mathbf{P}_s\pi\mathbf{P}_t^\top - \hat{\mathbf{P}}_s\pi\hat{\mathbf{P}}_t^\top\right|_{ij}\underbrace{\sum_{i,j}\pi_{ij}}_{=1}$$

$$= \max_{i,j}\left|\mathbf{P}_s\pi\mathbf{P}_t^\top - \hat{\mathbf{P}}_s\pi\hat{\mathbf{P}}_t^\top\right|_{ij}$$

$$\leq \max_{i,j}\left|\mathbf{P}_s\pi\mathbf{P}_t^\top - \mathbf{P}_s\pi\hat{\mathbf{P}}_t^\top\right|_{ij} + \max_{i,j}\left|\mathbf{P}_s\pi\hat{\mathbf{P}}_t^\top - \hat{\mathbf{P}}_s\pi\hat{\mathbf{P}}_t^\top\right|_{ij}$$

$$= \max_{i,j}\left|\sum_k(\mathbf{P}_s\pi)_{ik}(\mathbf{P}_t - \hat{\mathbf{P}}_t)_{jk}\right| + \max_{i,j}\left|\sum_k(\mathbf{P}_s - \hat{\mathbf{P}}_s)_{ik}(\pi\hat{\mathbf{P}}_t^\top)_{kj}\right|$$

$$\leq \|\mathbf{P}_t - \hat{\mathbf{P}}_t\|_{\infty,\infty}\max_i\sum_k|(\mathbf{P}_s\pi)|_{ik} + \|\hat{\mathbf{P}}_s - \mathbf{P}_s\|_{\infty,\infty}\max_j\sum_k|(\pi\hat{\mathbf{P}}_t^\top)_{kj}|$$

$$\leq \delta_2\,\max_i\sum_k|(\mathbf{P}_s\pi)|_{ik} + \delta_1\,\max_j\sum_k|(\pi\hat{\mathbf{P}}_t^\top)_{kj}|$$

$$\leq \delta_2\,\max_i\sum_k|\sum_l\mathbf{P}_{s,il}\pi_{lk}| + \delta_1\,\max_j\sum_k|\sum_l\pi_{kl}\mathbf{P}_{t,jl}|$$

$$\leq \delta_2\|\mathbf{P}_s\|_\infty\sum_{kl}\pi_{kl} + \delta_1\,\|\hat{\mathbf{P}}_t\|_\infty\sum_{kl}\pi_{kl} := \delta_2\|\mathbf{P}_s\|_\infty + \delta_1\,\|\tilde{\mathbf{P}}_t\|_\infty = \delta_1 + \delta_2\,.$$

Therefore, we conclude that

$$\|f - g\|_\infty \leq \|\mathbf{P}_s - \hat{\mathbf{P}}_s\|_{\infty,\infty} + \|\mathbf{P}_t - \hat{\mathbf{P}}_t\|_{\infty,\infty} := \Delta_{\mathrm{pert}}\,.$$

Now, from the definition of $f$ and $g$, we also have:

$$\nabla_\pi f(\pi) = -\left(\mathbf{P}_s \otimes \mathbf{P}_t + \mathbf{P}_s^\top \otimes \mathbf{P}_t^\top\right) \text{vec}(\pi) + \epsilon((1 + \log \pi_{ij}))_{i,j}$$

$$\nabla_\pi g(\pi) = -\left(\hat{\mathbf{P}}_s \otimes \hat{\mathbf{P}}_t + \hat{\mathbf{P}}_s^\top \otimes \hat{\mathbf{P}}_t^\top\right) \text{vec}(\pi) + \epsilon((1 + \log \pi_{ij}))_{i,j}$$

$$\nabla_\pi^2 f(\pi) = -\left(\mathbf{P}_s \otimes \mathbf{P}_t + \mathbf{P}_s^\top \otimes \mathbf{P}_t^\top\right) + \epsilon\text{diag}\left(\left\{\frac{1}{\pi_{ij}}\right\}\right)$$

$$\nabla_\pi^2 g(\pi) = -\left(\hat{\mathbf{P}}_s \otimes \hat{\mathbf{P}}_t + \hat{\mathbf{P}}_s^\top \otimes \hat{\mathbf{P}}_t^\top\right) + \epsilon\text{diag}\left(\left\{\frac{1}{\pi_{ij}}\right\}\right)$$

As a consequence, we have:

$$\|\nabla_\pi (g - f)(\pi)\|_F \le 2 \left\|(\mathbf{P}_s \otimes \mathbf{P}_t) - (\hat{\mathbf{P}}_s \otimes \hat{\mathbf{P}}_t)\right\|_{\text{op}} \|\pi\|_F .$$

Hence, we have:

$$\kappa = 2 \left\|(\mathbf{P}_s \otimes \mathbf{P}_t) - (\hat{\mathbf{P}}_s \otimes \hat{\mathbf{P}}_t)\right\|_{\text{op}} .$$

Now, from the Hessian, we know $\nabla_\pi^2 f(\pi^*) \succ 0$ if $\max_{ij} \pi_{ij}^* < \epsilon/(2\|\mathbf{P}_s \otimes \mathbf{P}_t\|_{\text{op}})$. This condition is met as per our assumption that

$$\|\pi^*\|_\infty \le \frac{\epsilon}{C_1\|\mathbf{P}_s \otimes \mathbf{P}_t\|_{\text{op}}} .$$

with $C_1 > 2$. Fix $C_2$ such that $(1/C_2) + (1/C_1) < (1/2)$, and set $\tau = \epsilon/(C_2\|\mathbf{P}_s \otimes \mathbf{P}_t\|_{\text{op}})$. Then, for any $\|\pi - \pi^*\|_F \le \tau$,

$$\max_{ij} \pi_{ij} \le \tau + \max_{ij} \pi_{ij}^* \le \frac{\epsilon}{C_2\|\mathbf{P}_s \otimes \mathbf{P}_t\|_{\text{op}}} + \frac{\epsilon}{C_1\|\mathbf{P}_s \otimes \mathbf{P}_t\|_{\text{op}}} < \frac{\epsilon}{2\|\mathbf{P}_s \otimes \mathbf{P}_t\|_{\text{op}}} .$$

In this neighborhood, we have:

$$\inf_{\pi \in B(\pi^*;\tau)} \inf_{\|v\|_2=1} v^\top \nabla_\pi^2 f(\pi) v \ge \left\{\left(\frac{1}{C_1} + \frac{1}{C_2}\right)^{-1} - 2\right\} \|\mathbf{P}_s \otimes \mathbf{P}_t\|_{\text{op}} := \frac{1}{C_3}\|\mathbf{P}_s \otimes \mathbf{P}_t\|_{\text{op}} := \frac{\mu}{2} .$$

Therefore, using Lemma B.2 we can conclude:

$$\|\hat{\pi} - \pi^*\|_F \le \frac{20\left\|(\mathbf{P}_s \otimes \mathbf{P}_t) - (\hat{\mathbf{P}}_s \otimes \hat{\mathbf{P}}_t)\right\|_{\text{op}}}{\|\mathbf{P}_s \otimes \mathbf{P}_t\|_{\text{op}}} \qquad \text{as soon as} \qquad \Delta_{\text{pert}} \le \frac{\tau^2\mu}{4} .$$

By our definition of $\delta$ and $\mu$, we have:

$$\frac{\tau^2\mu}{4} = \frac{\left(\frac{\epsilon}{C_2\|\mathbf{P}_s \otimes \mathbf{P}_t\|_{\text{op}}}\right)^2}{4} \cdot \frac{2\|\mathbf{P}_s \otimes \mathbf{P}_t\|_{\text{op}}}{C_3} .$$

Recall that our assumption is:

$$\Delta_{\text{pert}} = \|\mathbf{P}_s - \hat{\mathbf{P}}_s\|_{\infty,\infty} + \|\mathbf{P}_t - \hat{\mathbf{P}}_t\|_{\infty,\infty} \le \frac{\tau^2\mu}{4} .$$

Substituting the expression for $\tau^2\mu$:

$$\Delta_{\text{pert}} \le \frac{\epsilon^2}{2C_2^2 C_3(\|\mathbf{P}_s \otimes \mathbf{P}_t\|_{\text{op}})} .$$

This guarantees that the perturbed solution $\hat{\pi}$ remains in the neighborhood $B(\pi^*, \tau)$ of the true solution $\pi^*$. Thus, the stability of the optimal transport plan is ensured under perturbations of the graphon estimators, completing the proof.

# C  Proof of Theorem 4.2

*Proof.* The proof is simple, which depends on the estimator error of $\mathbf{P}_s$ and $\mathbf{P}_t$. Recall that:

$$\mathrm{GW}_2^2(\hat{\mathbf{P}}_s^{\mathrm{init}}, \hat{\mathbf{P}}_t^{\mathrm{init}}) = \inf_{\pi \in \Gamma_n} \sum_{i,i'=1}^{n_S} \sum_{j,j'=1}^{n_T} |\hat{\mathbf{P}}_{S,ii'}^{\mathrm{init}} - \hat{P}_{T,jj'}|^2 \pi_{ij} \pi_{i'j'}$$

$$\mathrm{GW}_2^2(\mathbf{P}_s, \mathbf{P}_t) = \inf_{\pi \in \Gamma_n} \sum_{i,i'=1}^{n_S} \sum_{j,j'=1}^{n_T} |P_{S,ii'} - P_{T,jj'}|^2 \pi_{ij} \pi_{i'j'}$$

where $\Gamma_n$ is the set of couplings, i.e.

$$\Gamma_n = \left\{ \pi : \sum_i \pi_{ij} = \frac{1}{n_T}, \quad \sum_j \pi_{ij} = \frac{1}{n_S} \right\}.$$

Now let $\hat{\pi}_n$ and $\pi_n$ denotes the minimizer/optimal coupling of $\mathrm{GW}_2^2(\hat{\mathbf{P}}_s^{\mathrm{init}}, \hat{\mathbf{P}}_t^{\mathrm{init}})$ and $\mathrm{GW}_2^2(\mathbf{P}_s, \mathbf{P}_t)$ respectively. Then we have:

$$\mathrm{GW}_2^2(\mathbf{P}_s, \mathbf{P}_t)$$
$$= \inf_{\pi \in \Gamma_n} \sum_{i,i'=1}^{n_S} \sum_{j,j'=1}^{n_T} |\mathbf{P}_{S,ii'} - \mathbf{P}_{T,jj'}|^2 \pi_{ij} \pi_{i'j'}$$
$$\leq \sum_{i,i'=1}^{n_S} \sum_{j,j'=1}^{n_T} |\mathbf{P}_{S,ii'} - \mathbf{P}_{T,jj'}|^2 \hat{\pi}_{n,ij} \hat{\pi}_{n,i'j'}$$
$$= \sum_{i,i'=1}^{n_S} \sum_{j,j'=1}^{n_T} |\mathbf{P}_{S,ii'} - \hat{\mathbf{P}}_{S,ii'}^{\mathrm{init}} + \hat{\mathbf{P}}_{S,ii'}^{\mathrm{init}} - \hat{\mathbf{P}}_{T,jj'}^{\mathrm{init}} + \hat{\mathbf{P}}_{T,jj'}^{\mathrm{init}} - \mathbf{P}_{T,jj'}|^2 \hat{\pi}_{n,ij} \hat{\pi}_{n,i'j'}$$
$$\leq 4 \left[ \sum_{i,i'=1}^{n_S} \sum_{j,j'=1}^{n_T} |\hat{\mathbf{P}}_{S,ii'}^{\mathrm{init}} - \mathbf{P}_{S,ii'}|^2 \hat{\pi}_{n,ij} \hat{\pi}_{n,i'j'} \right.$$
$$\left. + \sum_{i,i'=1}^{n_S} \sum_{j,j'=1}^{n_T} |\hat{\mathbf{P}}_{S,ii'}^{\mathrm{init}} - \hat{\mathbf{P}}_{T,jj'}^{\mathrm{init}}|^2 \hat{\pi}_{n,ij} \hat{\pi}_{n,i'j'} + \sum_{i,i'=1}^{n_S} \sum_{j,j'=1}^{n_T} |\mathbf{P}_{T,jj'} - \hat{\mathbf{P}}_{T,jj'}^{\mathrm{init}}|^2 \hat{\pi}_{n,ij} \hat{\pi}_{n,i'j'} \right]$$
$$\leq 4 \left[ \frac{1}{n_S^2} \sum_{i,i'=1}^{n_S} |\hat{\mathbf{P}}_{S,ii'}^{\mathrm{init}} - \mathbf{P}_{S,ii'}|^2 + \sum_{i,i'=1}^{n_S} \sum_{j,j'=1}^{n_T} |\hat{\mathbf{P}}_{S,ii'}^{\mathrm{init}} - \hat{\mathbf{P}}_{T,jj'}^{\mathrm{init}}|^2 \hat{\pi}_{n,ij} \hat{\pi}_{n,i'j'} + \frac{1}{n_T^2} \sum_{j,j'=1}^{n_T} |\mathbf{P}_{T,jj'} - \hat{\mathbf{P}}_{T,jj'}^{\mathrm{init}}|^2 \right]$$
$$\leq 4 \left( \delta_{n_S} + \mathrm{GW}_2^2(\hat{\mathbf{P}}_S^{\mathrm{init}}, \hat{\mathbf{P}}_T^{\mathrm{init}}) + \delta_{n_T} \right) = 4 \left( \delta_n + \mathrm{GW}_2^2(\hat{\mathbf{P}}_S^{\mathrm{init}}, \hat{\mathbf{P}}_T^{\mathrm{init}}) \right).$$

Similarly, for the other side:

$$\mathrm{GW}_2^2(\hat{\mathbf{P}}_s^{\mathrm{init}}, \hat{\mathbf{P}}_t^{\mathrm{init}})$$

$$= \inf_{\pi \in \Gamma_n} \sum_{i,i'=1}^{n_S} \sum_{j,j'=1}^{n_T} |\hat{\mathbf{P}}_{S,ii'}^{\mathrm{init}} - \hat{\mathbf{P}}_{T,jj'}^{\mathrm{init}}|^2 \pi_{ij} \pi_{i'j'}$$

$$\leq \sum_{i,i'=1}^{n_S} \sum_{j,j'=1}^{n_T} |\hat{\mathbf{P}}_{S,ii'}^{\mathrm{init}} - \hat{\mathbf{P}}_{T,jj'}^{\mathrm{init}}|^2 \pi_{n,ij} \pi_{n,i'j'}$$

$$= \sum_{i,i'=1}^{n_S} \sum_{j,j'=1}^{n_T} |\hat{\mathbf{P}}_{S,ii'}^{\mathrm{init}} - \mathbf{P}_{S,ii'} + \mathbf{P}_{S,ii'} - \mathbf{P}_{T,jj'} + \mathbf{P}_{T,jj'} - \hat{\mathbf{P}}_{T,jj'}^{\mathrm{init}}|^2 \pi_{n,ij} \pi_{n,i'j'}$$

$$\leq 4 \left[ \sum_{i,i'=1}^{n_S} \sum_{j,j'=1}^{n_T} |\hat{\mathbf{P}}_{S,ii'}^{\mathrm{init}} - \mathbf{P}_{S,ii'}|^2 \pi_{n,ij} \pi_{n,i'j'} \right.$$

$$\left. + \sum_{i,i'=1}^{n_S} \sum_{j,j'=1}^{n_T} |\mathbf{P}_{S,ii'} - \mathbf{P}_{T,jj'}|^2 \pi_{n,ij} \pi_{n,i'j'} + \sum_{i,i'=1}^{n_S} \sum_{j,j'=1}^{n_T} |\mathbf{P}_{T,jj'} - \hat{\mathbf{P}}_{T,jj'}^{\mathrm{init}}|^2 \pi_{n,ij} \pi_{n,i'j'} \right]$$

$$\leq 4 \left[ \frac{1}{n_S^2} \sum_{i,i'=1}^{n_S} |\hat{\mathbf{P}}_{S,ii'}^{\mathrm{init}} - \mathbf{P}_{S,ii'}|^2 + \sum_{i,i'=1}^{n_S} \sum_{j,j'=1}^{n_T} |\mathbf{P}_{S,ii'} - \mathbf{P}_{T,jj'}|^2 \pi_{n,ij} \pi_{n,i'j'} + \frac{1}{n_T^2} \sum_{j,j'=1}^{n_T} |\mathbf{P}_{T,jj'} - \hat{\mathbf{P}}_{T,jj'}^{\mathrm{init}}|^2 \right]$$

$$\leq 4 \left( \delta_{n_S} + \mathrm{GW}_2^2(\mathbf{P}_s, \mathbf{P}_t) + \delta_{n_T} \right) = 4 \left( \delta_n + \mathrm{GW}_2^2(\mathbf{P}_s, \mathbf{P}_t) \right) .$$

$\square$

## D    Proof of Theorem 4.3

*Proof.* We start with the definition of the GW distance:

$$\mathrm{GW}_2^2(f, g) = \int \int (f(x, x') - g(y, y'))^2 \, d\pi^*(x, y) \, d\pi^*(x', y')$$

$$= \int f^2(x, x') \, dx \, dx' + \int g^2(y, y') \, dy \, dy' - 2 \int \int f(x, x')g(y, y') d\pi^*(x, y) \, d\pi^*(x', y')$$

$$= \int f^2(x, x') \, dx \, dx' + \int g^2(y, y') \, dy \, dy' - 2 \int \int \tilde{g}(y, y')g(y, y') \, dy \, dy'$$

Furthermore, we have:

$$\int \tilde{g}^2(y, y') \, dy \, dy' = \int \left( \int_{[0,1]^2} \pi^*(y, x) f(x, x') \pi^*(x', y) \, dx \, dx' \right)^2 \, dy dy'$$

$$\leq \int \int f^2(x, x') \pi^*(y, x) \pi^*(x', y) \, dx dx' dy dy'$$

$$= \int f^2(x, x') \, dx \, dx' .$$

Therefore, we have:

$$\|\tilde{g} - g\|_2^2 = \int_{[0,1]^2} (\tilde{g}(y, y') - g(y, y'))^2 \, dy dy'$$

$$= \int \tilde{g}^2(y, y') \, dy dy' + \int g^2(y, y') \, dy dy' - 2 \int \tilde{g}(y, y')g(y, y') \, dy dy'$$

$$\leq \int f^2(x, x') \, dx dx' + \int g^2(y, y') \, dy dy' - 2 \int \int f(x, x')g(y, y') \, d\pi^*(x, y) d\pi^*(x', y')$$

$$= \mathrm{GW}_2^2(f, g) .$$

This completes the proof.

$\square$

# E    Additional Simulation Results

## E.1    Additional results for Impact of Density Shift

To evaluate the robustness of GTRANS under density discrepancies between the source and target domains, we introduce a density shift parameter $\lambda \in [-0.5, 0.5]$ to simulate structured perturbations, defined as $f_s(u, v) = f_t(u, v) + \xi$, where $\xi \sim U(0, \lambda)$ if $\lambda > 0$ and $\xi \sim U(\lambda, 0)$ if $\lambda < 0$. We investigate the performance of GTRANS-GW and GTRANS-EGW across 10 different graphon functions (IDs: 1 to 10) with a fixed target sample size of 50 and a source sample size of 200. Each experiment is repeated 50 times to evaluate the adaptability of the models under varying levels of density shift. The results are summarized in Figure E1.

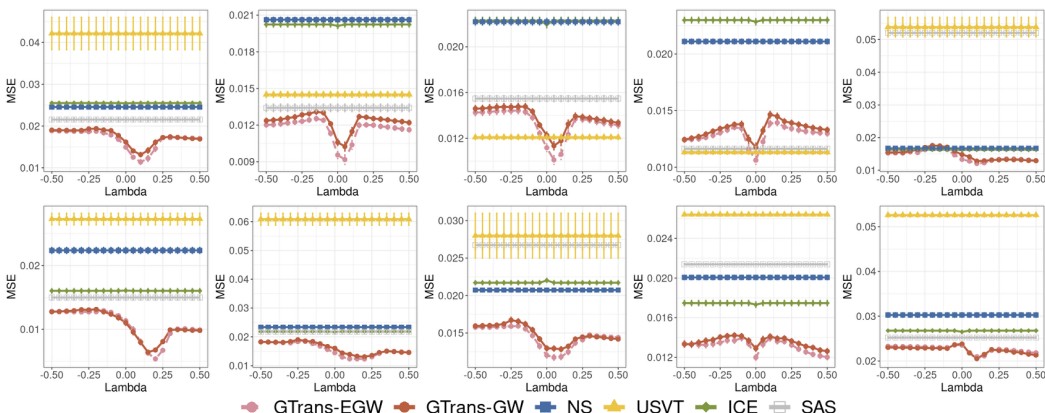

Figure E1: MSE performance of five methods as the density shift parameter $\lambda$ varies from $-0.5$ to $0.5$, with error bars representing $\pm 0.1$ standard deviations. GTRANS-GW (red circles with solid line), GTRANS-EGW (pink circles with dashed line), NS (blue squares with solid line), USVT (yellow triangles with solid line), ICE (green diamonds with solid line), SAS (gray hollow squares with solid line). Top row: graphons 1-5; bottom row: graphons 5-10.

Figure E1 shows the average MSE of different methods as the density shift parameter $\lambda$ varies from $-0.5$ to $0.5$. Both **GTRANS-GW (red solid line)** and **GTRANS-EGW (pink dashed line)** demonstrate substantial improvements over the baseline methods (NS, USVT, ICE, and SAS), which remain constant across all $\lambda$ values. Notably, as $\lambda$ shifts from negative to positive, GTRANS methods effectively adjust their alignment, capturing structural changes and minimizing MSE. This result underscores the capability of GTRANS-GW and GTRANS-EGW to dynamically adapt to source-target discrepancies, achieving consistently lower MSE across all scenarios. Also, we notice that for dense graphons like Graphon 2 (density = 0.5) and Graphon 9 (density = 0.5), GTRANS-GW and GTRANS-EGW exhibit symmetric U-shaped MSE curves around zero, indicating balanced performance regardless of source density. In contrast, for sparse graphons like Graphon 6 (density = 0.13) and Graphon 10 (density = 0.19), the lowest MSE is achieved with a slightly denser source, highlighting the need for stronger initial signals when the target is sparse. Unique structural patterns in Graphon 5 (anti-diagonal) and Graphon 7 (oscillatory) cause non-monotonic fluctuations under density shift, reflecting sensitivity to structural misalignments.

## E.2    Additional Simulation for Transfer Between Different Graphons

We conduct simulations across ten distinct graphon structures, indexed from 1 to 10. For each graphon, we generate a target domain with a fixed sample size of $n_t = 50$ and a source domain with $n_s = 500$, simulating realistic structural perturbations by adding uniform noise sampled from $[-0.01, 0.01]$. The evaluation includes USVT, ICE, and SAS, and our proposed transfer learning frameworks: GTRANS-GW and GTRANS-EGW ($\epsilon = 0.01$). For comprehensive analysis, we compute the Mean Squared Error (MSE) for each method and aggregate the results.

Table E2: Comparison of different methods for various transfer scenarios. Best-performing methods are **bolded**.

| Scenario | GTRANS-GW | GTRANS-EGW | NS | USVT | ICE | SAS |
|---|---|---|---|---|---|---|
| $1 \rightarrow 2$ | $1.4 \pm 0.3$ | $\mathbf{1.3 \pm 0.3}$ | $2.1 \pm 0.2$ | $1.5 \pm 0.3$ | $2.1 \pm 0.2$ | $1.4 \pm 0.4$ |
| $1 \rightarrow 3$ | $1.6 \pm 0.3$ | $1.5 \pm 0.3$ | $2.3 \pm 0.3$ | $\mathbf{1.2 \pm 0.2}$ | $2.2 \pm 0.3$ | $1.6 \pm 0.3$ |
| $1 \rightarrow 4$ | $1.3 \pm 0.1$ | $1.3 \pm 0.2$ | $2.1 \pm 0.2$ | $\mathbf{1.0 \pm 0.1}$ | $2.3 \pm 0.2$ | $1.1 \pm 0.2$ |
| $1 \rightarrow 5$ | $1.6 \pm 0.2$ | $\mathbf{1.5 \pm 0.2}$ | $1.7 \pm 0.1$ | $3.5 \pm 1.5$ | $1.7 \pm 0.2$ | $5.5 \pm 1.0$ |
| $1 \rightarrow 6$ | $\mathbf{1.0 \pm 0.2}$ | $1.1 \pm 0.2$ | $2.1 \pm 0.3$ | $3.0 \pm 0.9$ | $1.7 \pm 0.3$ | $1.6 \pm 0.4$ |
| $1 \rightarrow 7$ | $1.1 \pm 0.4$ | $\mathbf{0.9 \pm 0.4}$ | $2.3 \pm 0.3$ | $5.9 \pm 2.7$ | $2.2 \pm 0.3$ | $2.4 \pm 0.6$ |
| $1 \rightarrow 8$ | $1.5 \pm 0.3$ | $\mathbf{1.4 \pm 0.3}$ | $2.0 \pm 0.2$ | $1.5 \pm 0.2$ | $2.1 \pm 0.2$ | $2.8 \pm 0.7$ |
| $1 \rightarrow 9$ | $1.8 \pm 0.4$ | $\mathbf{1.7 \pm 0.3}$ | $2.0 \pm 0.2$ | $2.6 \pm 0.1$ | $1.7 \pm 0.2$ | $2.1 \pm 0.1$ |
| $1 \rightarrow 10$ | $\mathbf{2.2 \pm 0.3}$ | $\mathbf{2.2 \pm 0.3}$ | $3.1 \pm 0.6$ | $5.3 \pm 0.6$ | $2.7 \pm 0.3$ | $2.6 \pm 0.3$ |
| $2 \rightarrow 1$ | $1.9 \pm 0.2$ | $\mathbf{1.8 \pm 0.2}$ | $2.5 \pm 0.3$ | $5.1 \pm 4.6$ | $2.7 \pm 0.3$ | $2.1 \pm 0.3$ |
| $2 \rightarrow 3$ | $1.6 \pm 0.3$ | $1.6 \pm 0.3$ | $2.3 \pm 0.3$ | $\mathbf{1.2 \pm 0.2}$ | $2.2 \pm 0.3$ | $1.6 \pm 0.3$ |
| $2 \rightarrow 4$ | $1.3 \pm 0.2$ | $1.3 \pm 0.2$ | $2.1 \pm 0.2$ | $\mathbf{1.0 \pm 0.1}$ | $2.3 \pm 0.2$ | $1.1 \pm 0.2$ |
| $2 \rightarrow 5$ | $1.6 \pm 0.2$ | $\mathbf{1.5 \pm 0.2}$ | $1.7 \pm 0.1$ | $3.5 \pm 1.5$ | $1.7 \pm 0.2$ | $5.5 \pm 1.0$ |
| $2 \rightarrow 6$ | $\mathbf{1.0 \pm 0.2}$ | $\mathbf{1.0 \pm 0.2}$ | $2.1 \pm 0.3$ | $3.0 \pm 0.9$ | $1.7 \pm 0.3$ | $1.6 \pm 0.4$ |
| $2 \rightarrow 7$ | $\mathbf{1.6 \pm 0.2}$ | $\mathbf{1.6 \pm 0.2}$ | $2.3 \pm 0.3$ | $5.9 \pm 2.7$ | $2.2 \pm 0.3$ | $2.4 \pm 0.6$ |
| $2 \rightarrow 8$ | $\mathbf{1.5 \pm 0.2}$ | $\mathbf{1.5 \pm 0.2}$ | $2.0 \pm 0.2$ | $\mathbf{1.5 \pm 0.2}$ | $2.1 \pm 0.2$ | $2.8 \pm 0.7$ |
| $2 \rightarrow 9$ | $1.8 \pm 0.5$ | $\mathbf{1.6 \pm 0.4}$ | $2.0 \pm 0.2$ | $2.6 \pm 0.1$ | $1.7 \pm 0.2$ | $2.1 \pm 0.1$ |
| $2 \rightarrow 10$ | $\mathbf{2.2 \pm 0.3}$ | $\mathbf{2.2 \pm 0.3}$ | $3.1 \pm 0.6$ | $5.3 \pm 0.6$ | $2.7 \pm 0.3$ | $2.6 \pm 0.3$ |
| $3 \rightarrow 1$ | $\mathbf{1.6 \pm 0.5}$ | $1.7 \pm 0.2$ | $2.5 \pm 0.3$ | $5.1 \pm 4.6$ | $2.7 \pm 0.3$ | $2.1 \pm 0.3$ |
| $3 \rightarrow 2$ | $1.5 \pm 0.3$ | $\mathbf{1.4 \pm 0.3}$ | $2.1 \pm 0.2$ | $1.5 \pm 0.3$ | $2.1 \pm 0.2$ | $1.4 \pm 0.4$ |
| $3 \rightarrow 4$ | $1.4 \pm 0.2$ | $1.3 \pm 0.2$ | $2.1 \pm 0.2$ | $\mathbf{1.0 \pm 0.1}$ | $2.3 \pm 0.2$ | $1.1 \pm 0.2$ |
| $3 \rightarrow 5$ | $\mathbf{1.4 \pm 0.2}$ | $\mathbf{1.4 \pm 0.2}$ | $1.7 \pm 0.1$ | $3.5 \pm 1.5$ | $1.7 \pm 0.2$ | $5.5 \pm 1.0$ |
| $3 \rightarrow 6$ | $\mathbf{0.9 \pm 0.2}$ | $1.0 \pm 0.2$ | $2.1 \pm 0.3$ | $3.0 \pm 0.9$ | $1.7 \pm 0.3$ | $1.6 \pm 0.4$ |
| $3 \rightarrow 7$ | $\mathbf{1.5 \pm 0.2}$ | $\mathbf{1.5 \pm 0.2}$ | $2.3 \pm 0.3$ | $5.9 \pm 2.7$ | $2.2 \pm 0.3$ | $2.4 \pm 0.6$ |
| $3 \rightarrow 8$ | $1.5 \pm 0.3$ | $\mathbf{1.4 \pm 0.3}$ | $2.0 \pm 0.2$ | $1.5 \pm 0.2$ | $2.1 \pm 0.2$ | $2.8 \pm 0.7$ |
| $3 \rightarrow 9$ | $1.9 \pm 0.3$ | $1.8 \pm 0.3$ | $2.0 \pm 0.2$ | $2.6 \pm 0.1$ | $\mathbf{1.7 \pm 0.2}$ | $2.1 \pm 0.1$ |
| $3 \rightarrow 10$ | $2.2 \pm 0.3$ | $\mathbf{2.1 \pm 0.3}$ | $3.1 \pm 0.6$ | $5.3 \pm 0.6$ | $2.7 \pm 0.3$ | $2.6 \pm 0.3$ |
| $4 \rightarrow 1$ | $\mathbf{1.7 \pm 0.2}$ | $\mathbf{1.7 \pm 0.2}$ | $2.5 \pm 0.3$ | $5.1 \pm 4.6$ | $2.7 \pm 0.3$ | $2.1 \pm 0.3$ |
| $4 \rightarrow 2$ | $\mathbf{1.3 \pm 0.3}$ | $\mathbf{1.3 \pm 0.3}$ | $2.1 \pm 0.2$ | $1.5 \pm 0.3$ | $2.1 \pm 0.2$ | $1.4 \pm 0.4$ |
| $4 \rightarrow 3$ | $1.4 \pm 0.3$ | $1.3 \pm 0.3$ | $2.3 \pm 0.3$ | $\mathbf{1.2 \pm 0.2}$ | $2.2 \pm 0.3$ | $1.6 \pm 0.3$ |
| $4 \rightarrow 5$ | $\mathbf{1.3 \pm 0.2}$ | $\mathbf{1.3 \pm 0.2}$ | $1.7 \pm 0.1$ | $3.5 \pm 1.5$ | $1.7 \pm 0.2$ | $5.5 \pm 1.0$ |
| $4 \rightarrow 6$ | $\mathbf{1.0 \pm 0.2}$ | $1.1 \pm 0.2$ | $2.1 \pm 0.3$ | $3.0 \pm 0.9$ | $1.7 \pm 0.3$ | $1.6 \pm 0.4$ |
| $4 \rightarrow 7$ | $\mathbf{1.5 \pm 0.2}$ | $\mathbf{1.5 \pm 0.2}$ | $2.3 \pm 0.3$ | $5.9 \pm 2.7$ | $2.2 \pm 0.3$ | $2.4 \pm 0.6$ |
| $4 \rightarrow 8$ | $\mathbf{1.4 \pm 0.2}$ | $\mathbf{1.4 \pm 0.2}$ | $2.0 \pm 0.2$ | $1.5 \pm 0.2$ | $2.1 \pm 0.2$ | $2.8 \pm 0.7$ |
| $4 \rightarrow 9$ | $1.5 \pm 0.3$ | $\mathbf{1.4 \pm 0.3}$ | $2.0 \pm 0.2$ | $2.6 \pm 0.1$ | $1.7 \pm 0.2$ | $2.1 \pm 0.1$ |
| $4 \rightarrow 10$ | $2.2 \pm 0.3$ | $\mathbf{2.1 \pm 0.3}$ | $3.1 \pm 0.6$ | $5.3 \pm 0.6$ | $2.7 \pm 0.3$ | $2.6 \pm 0.3$ |
| $5 \rightarrow 1$ | $\mathbf{2.0 \pm 0.3}$ | $2.1 \pm 0.3$ | $2.5 \pm 0.3$ | $5.1 \pm 4.6$ | $2.6 \pm 0.2$ | $2.1 \pm 0.3$ |
| $5 \rightarrow 2$ | $1.7 \pm 0.3$ | $1.6 \pm 0.3$ | $2.1 \pm 0.2$ | $1.5 \pm 0.3$ | $2.1 \pm 0.2$ | $\mathbf{1.4 \pm 0.4}$ |
| $5 \rightarrow 3$ | $1.7 \pm 0.3$ | $1.7 \pm 0.3$ | $2.3 \pm 0.3$ | $\mathbf{1.2 \pm 0.2}$ | $2.2 \pm 0.3$ | $1.6 \pm 0.3$ |
| $5 \rightarrow 4$ | $1.4 \pm 0.1$ | $1.5 \pm 0.2$ | $2.1 \pm 0.2$ | $\mathbf{1.0 \pm 0.1}$ | $2.3 \pm 0.2$ | $1.1 \pm 0.2$ |
| $5 \rightarrow 6$ | $\mathbf{1.2 \pm 0.2}$ | $\mathbf{1.2 \pm 0.2}$ | $2.1 \pm 0.3$ | $3.0 \pm 0.9$ | $1.7 \pm 0.3$ | $1.6 \pm 0.4$ |
| $5 \rightarrow 7$ | $\mathbf{1.8 \pm 0.3}$ | $\mathbf{1.8 \pm 0.3}$ | $2.3 \pm 0.3$ | $5.9 \pm 2.7$ | $2.2 \pm 0.2$ | $2.4 \pm 0.6$ |
| $5 \rightarrow 8$ | $\mathbf{1.5 \pm 0.2}$ | $1.7 \pm 0.2$ | $2.0 \pm 0.2$ | $\mathbf{1.5 \pm 0.2}$ | $2.1 \pm 0.2$ | $2.8 \pm 0.7$ |
| $5 \rightarrow 9$ | $2.1 \pm 0.3$ | $\mathbf{2.0 \pm 0.3}$ | $\mathbf{2.0 \pm 0.2}$ | $2.6 \pm 0.1$ | $1.8 \pm 0.3$ | $2.1 \pm 0.1$ |
| $5 \rightarrow 10$ | $2.6 \pm 0.3$ | $\mathbf{2.5 \pm 0.3}$ | $3.1 \pm 0.6$ | $5.3 \pm 0.6$ | $2.8 \pm 0.3$ | $2.6 \pm 0.3$ |
| $6 \rightarrow 1$ | $\mathbf{2.0 \pm 0.3}$ | $\mathbf{2.0 \pm 0.3}$ | $2.5 \pm 0.3$ | $5.1 \pm 4.6$ | $2.7 \pm 0.3$ | $2.1 \pm 0.3$ |
| $6 \rightarrow 2$ | $\mathbf{1.3 \pm 0.3}$ | $\mathbf{1.3 \pm 0.3}$ | $2.1 \pm 0.2$ | $1.5 \pm 0.3$ | $2.1 \pm 0.2$ | $1.4 \pm 0.4$ |
| $6 \rightarrow 3$ | $1.5 \pm 0.3$ | $1.5 \pm 0.3$ | $2.3 \pm 0.3$ | $\mathbf{1.2 \pm 0.2}$ | $2.2 \pm 0.3$ | $1.6 \pm 0.3$ |
| $6 \rightarrow 4$ | $1.2 \pm 0.2$ | $1.2 \pm 0.2$ | $2.1 \pm 0.2$ | $\mathbf{1.0 \pm 0.1}$ | $2.3 \pm 0.2$ | $1.1 \pm 0.2$ |
| $6 \rightarrow 5$ | $1.6 \pm 0.2$ | $\mathbf{1.5 \pm 0.2}$ | $1.7 \pm 0.1$ | $3.5 \pm 1.5$ | $1.7 \pm 0.2$ | $5.5 \pm 1.0$ |

| Scenario | GTRANS-GW | GTRANS-EGW | NS | USVT | ICE | SAS |
|---|---|---|---|---|---|---|
| $6 \rightarrow 7$ | **1.8 ± 0.3** | **1.8 ± 0.3** | 2.3 ± 0.3 | 5.9 ± 2.7 | 2.2 ± 0.3 | 2.4 ± 0.6 |
| $6 \rightarrow 8$ | 1.6 ± 0.2 | **1.5 ± 0.2** | 2.0 ± 0.2 | **1.5 ± 0.2** | 2.1 ± 0.2 | 2.8 ± 0.7 |
| $6 \rightarrow 9$ | **1.5 ± 0.4** | 1.6 ± 0.4 | 2.0 ± 0.2 | 2.6 ± 0.1 | 1.7 ± 0.2 | 2.1 ± 0.1 |
| $6 \rightarrow 10$ | **2.3 ± 0.3** | **2.3 ± 0.3** | 3.1 ± 0.6 | 5.3 ± 0.6 | 2.7 ± 0.3 | 2.6 ± 0.3 |
| $7 \rightarrow 1$ | 2.0 ± 0.3 | **1.9 ± 0.3** | 2.5 ± 0.3 | 5.1 ± 4.6 | 2.7 ± 0.3 | 2.1 ± 0.3 |
| $7 \rightarrow 2$ | **1.3 ± 0.3** | **1.3 ± 0.3** | 2.1 ± 0.2 | 1.5 ± 0.3 | 2.1 ± 0.2 | 1.4 ± 0.4 |
| $7 \rightarrow 3$ | 1.6 ± 0.3 | 1.5 ± 0.3 | 2.3 ± 0.3 | **1.2 ± 0.2** | 2.2 ± 0.3 | 1.6 ± 0.3 |
| $7 \rightarrow 4$ | 1.3 ± 0.2 | 1.2 ± 0.2 | 2.1 ± 0.2 | **1.0 ± 0.1** | 2.3 ± 0.2 | 1.1 ± 0.2 |
| $7 \rightarrow 5$ | 1.6 ± 0.2 | **1.5 ± 0.2** | 1.7 ± 0.1 | 3.5 ± 1.5 | 1.7 ± 0.2 | 5.5 ± 1.0 |
| $7 \rightarrow 6$ | **0.9 ± 0.2** | 1.0 ± 0.3 | 2.1 ± 0.3 | 3.0 ± 0.9 | 1.7 ± 0.3 | 1.6 ± 0.4 |
| $7 \rightarrow 8$ | 1.6 ± 0.2 | 1.6 ± 0.2 | 2.0 ± 0.2 | **1.5 ± 0.2** | 2.1 ± 0.2 | 2.8 ± 0.7 |
| $7 \rightarrow 9$ | 1.7 ± 0.4 | 1.7 ± 0.4 | 2.0 ± 0.2 | 2.6 ± 0.1 | **1.7 ± 0.2** | 2.1 ± 0.1 |
| $7 \rightarrow 10$ | 2.3 ± 0.4 | **2.2 ± 0.3** | 3.1 ± 0.6 | 5.3 ± 0.6 | 2.7 ± 0.3 | 2.6 ± 0.3 |
| $8 \rightarrow 1$ | 1.8 ± 0.3 | **1.5 ± 0.3** | 2.5 ± 0.3 | 5.1 ± 4.6 | 2.7 ± 0.3 | 2.1 ± 0.3 |
| $8 \rightarrow 2$ | **1.4 ± 0.3** | **1.4 ± 0.3** | 2.1 ± 0.2 | 1.5 ± 0.3 | 2.1 ± 0.2 | 1.4 ± 0.4 |
| $8 \rightarrow 3$ | 1.7 ± 0.3 | 1.6 ± 0.3 | 2.3 ± 0.3 | **1.2 ± 0.2** | 2.2 ± 0.3 | 1.6 ± 0.3 |
| $8 \rightarrow 4$ | 1.3 ± 0.2 | 1.3 ± 0.2 | 2.1 ± 0.2 | **1.0 ± 0.1** | 2.3 ± 0.2 | 1.1 ± 0.2 |
| $8 \rightarrow 5$ | 1.6 ± 0.2 | **1.5 ± 0.2** | 1.7 ± 0.1 | 3.5 ± 1.5 | 1.7 ± 0.2 | 5.5 ± 1.0 |
| $8 \rightarrow 6$ | **1.0 ± 0.2** | **1.0 ± 0.2** | 2.1 ± 0.3 | 3.0 ± 0.9 | 1.7 ± 0.3 | 1.6 ± 0.4 |
| $8 \rightarrow 7$ | **1.3 ± 0.3** | 1.4 ± 0.3 | 2.3 ± 0.3 | 5.9 ± 2.7 | 2.2 ± 0.3 | 2.4 ± 0.6 |
| $8 \rightarrow 9$ | 1.9 ± 0.4 | 1.9 ± 0.3 | 2.0 ± 0.2 | 2.6 ± 0.1 | **1.7 ± 0.2** | 2.1 ± 0.1 |
| $8 \rightarrow 10$ | **2.3 ± 0.3** | **2.3 ± 0.3** | 3.1 ± 0.6 | 5.3 ± 0.6 | 2.7 ± 0.3 | 2.6 ± 0.3 |
| $9 \rightarrow 1$ | 2.4 ± 0.3 | 2.3 ± 0.2 | 2.5 ± 0.3 | 5.1 ± 4.6 | 2.7 ± 0.3 | **2.1 ± 0.3** |
| $9 \rightarrow 2$ | 1.5 ± 0.3 | 1.5 ± 0.3 | 2.1 ± 0.2 | 1.5 ± 0.3 | 2.1 ± 0.2 | **1.4 ± 0.4** |
| $9 \rightarrow 3$ | 1.9 ± 0.3 | 1.6 ± 0.3 | 2.3 ± 0.3 | **1.2 ± 0.2** | 2.2 ± 0.3 | 1.6 ± 0.3 |
| $9 \rightarrow 4$ | 1.6 ± 0.2 | 1.6 ± 0.2 | 2.1 ± 0.2 | **1.0 ± 0.1** | 2.3 ± 0.2 | 1.1 ± 0.2 |
| $9 \rightarrow 5$ | 1.6 ± 0.2 | **1.3 ± 0.2** | 1.7 ± 0.1 | 3.5 ± 1.5 | 1.7 ± 0.2 | 5.5 ± 1.0 |
| $9 \rightarrow 6$ | **1.2 ± 0.2** | **1.2 ± 0.2** | 2.1 ± 0.3 | 3.0 ± 0.9 | 1.7 ± 0.3 | 1.6 ± 0.4 |
| $9 \rightarrow 7$ | **1.9 ± 0.2** | 1.9 ± 0.3 | 2.3 ± 0.3 | 5.9 ± 2.7 | 2.2 ± 0.3 | 2.4 ± 0.6 |
| $9 \rightarrow 8$ | 2.0 ± 0.2 | 1.9 ± 0.2 | 2.0 ± 0.2 | **1.5 ± 0.2** | 2.1 ± 0.2 | 2.8 ± 0.7 |
| $9 \rightarrow 10$ | **2.2 ± 0.3** | 2.3 ± 0.3 | 3.1 ± 0.6 | 5.3 ± 0.6 | 2.7 ± 0.3 | 2.6 ± 0.3 |
| $10 \rightarrow 1$ | 2.1 ± 0.3 | **2.0 ± 0.3** | 2.5 ± 0.3 | 5.1 ± 4.6 | 2.7 ± 0.3 | 2.1 ± 0.3 |
| $10 \rightarrow 2$ | 1.3 ± 0.3 | **1.2 ± 0.3** | 2.1 ± 0.2 | 1.5 ± 0.3 | 2.1 ± 0.2 | 1.4 ± 0.4 |
| $10 \rightarrow 3$ | 1.6 ± 0.3 | 1.5 ± 0.3 | 2.3 ± 0.3 | **1.2 ± 0.2** | 2.2 ± 0.3 | 1.6 ± 0.3 |
| $10 \rightarrow 4$ | 1.2 ± 0.2 | 1.1 ± 0.2 | 2.1 ± 0.2 | **1.0 ± 0.1** | 2.3 ± 0.2 | 1.1 ± 0.2 |
| $10 \rightarrow 5$ | 1.6 ± 0.2 | **1.5 ± 0.2** | 1.7 ± 0.1 | 3.5 ± 1.5 | 1.7 ± 0.2 | 5.5 ± 1.0 |
| $10 \rightarrow 6$ | **1.3 ± 0.4** | 1.4 ± 0.3 | 2.1 ± 0.3 | 3.0 ± 0.9 | 1.7 ± 0.3 | 1.6 ± 0.4 |
| $10 \rightarrow 7$ | 1.9 ± 0.3 | **1.8 ± 0.3** | 2.3 ± 0.3 | 5.9 ± 2.7 | 2.2 ± 0.3 | 2.4 ± 0.6 |
| $10 \rightarrow 8$ | 1.6 ± 0.2 | **1.5 ± 0.2** | 2.0 ± 0.2 | **1.5 ± 0.2** | 2.1 ± 0.2 | 2.8 ± 0.7 |
| $10 \rightarrow 9$ | **1.5 ± 0.3** | 1.5 ± 0.4 | 2.0 ± 0.2 | 2.6 ± 0.1 | 1.7 ± 0.2 | 2.1 ± 0.1 |

### E.3 Ablation Study

Figure E2 presents the average estimation errors (MSE) of four configurations: **GTRANS** (red diamond with solid line), **GTRANS-NonDebias** (purple circle with dotted line), **GTRANS-NonSmooth** (orange triangle with dashed line), and **GTRANS-Adj** (yellow square with dash-dotted line). The x-axis represents the source sample size, and the y-axis shows the average MSE over 50 runs.

The four configurations are derived as follows: **GTRANS**: This is the full version, including all three steps from Algorithm 1. **GTRANS-NonDebias**: This variant removes the debiasing step (Step 3), directly using the transferred estimator $\hat{\mathbf{P}}_t^{trans2}$ as the final output. **GTRANS-NonSmooth**: This variant omits the neighborhood smoothing steps in the initial estimation (Step 1). **GTRANS-Adj**: This variant replaces the neighborhood smoothing (**Step 1**) entirely with raw adjacency matrices $A_s$ and $A_t$. In this setting, the optimal transport step (**Step 2**) operates directly on the unprocessed adjacency representations.

From the results, we observe that the complete **GTRANS** consistently outperforms its ablated variants across most graphon structures. The absence of smoothing (**GTRANS-NonSmooth**) introduces sharp oscillations in MSE, indicating the sensitivity to noise. **GTRANS-NonDebias** struggles to correct for transfer-induced discrepancies, especially when source-target alignment is imperfect. **GTRANS-Adj** shows consistently higher MSE, highlighting that raw adjacency matrices lack smoothness for effective GW-based alignment. This underscores the necessity of the smoothing step in GTRANS to achieve consistent and robust estimations. Therefore, the debiasing and smoothing mechanisms in GTRANS are not just additive; they are essential for robustness.

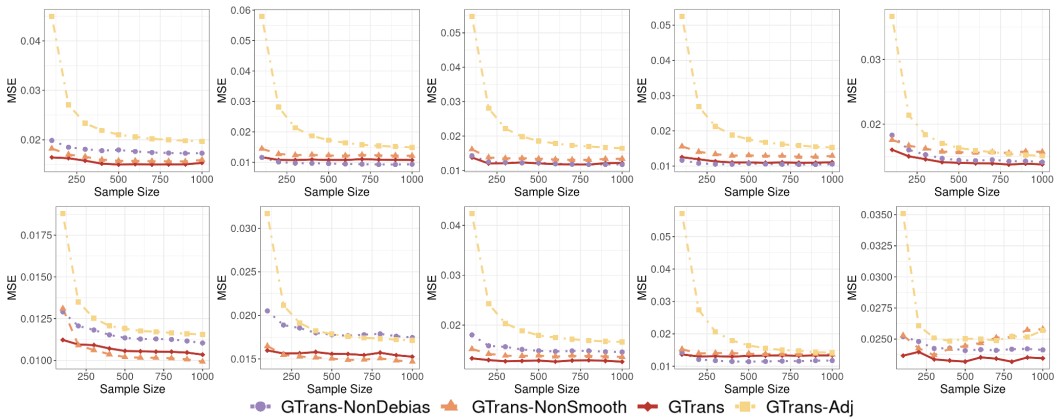

(a) Ablation Study with Increasing Source Sample Size.

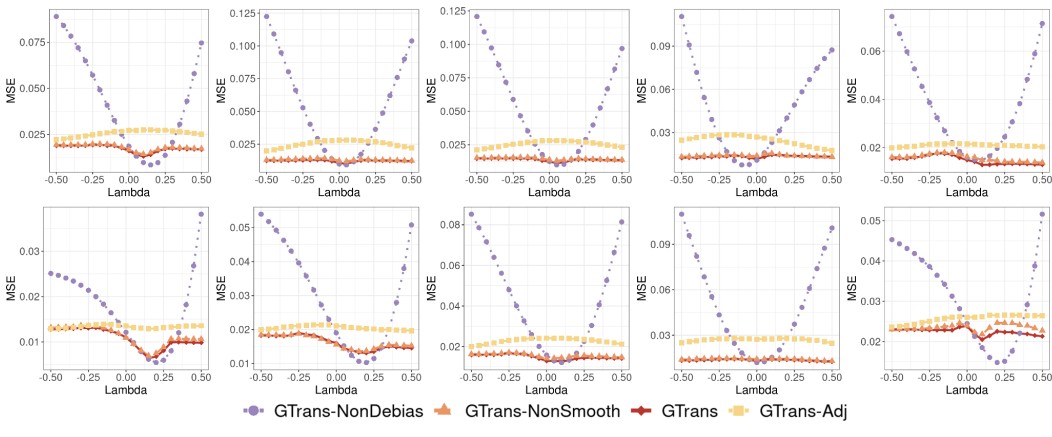

(b) Ablation Study with Varying $\lambda$.

Figure E2: Ablation study of different estimation methods: (a) illustrates the impact of increasing source sample size on estimation accuracy, and (b) demonstrates the effect of varying density shift ($\lambda$) on the performance of GTRANS, GTRANS-NonDebias, GTRANS-NonSmooth, and GTRANS-Adj.

### E.4 Hyperparameter Selection

#### E.4.1 Debiasing Threshold $\delta$ Selection

**Threshold $\delta$.** We evaluate the stability of the selected threshold $\delta^*$ under matched source and target distributions. Fixing the target size at $n_t = 50$, we vary the source size in $\{100, 200, 400, 800\}$ and generate both graphs from the same graphon (`graphon_id` $\in \{1, \ldots, 10\}$). Source perturbations are introduced via a structural noise parameter $\lambda \in [-0.5, 0.5]$. Thresholds $\delta \in \{0.01, 0.02, \ldots, 0.50\}$ are evaluated using the mean squared error (MSE) between the estimated and true target graphons. We evaluate a fixed threshold $\delta = 0.15$ across various source perturbations. If it aligns with the side of the optimal jump point determined by the GW distance $d$, it is selected; otherwise, the best

threshold on that side is chosen. Empirically, $\delta = 0.15$ remains robust, while under EGW with $\epsilon = 0.01$, the optimal choice converges to $\delta = 0.18$.

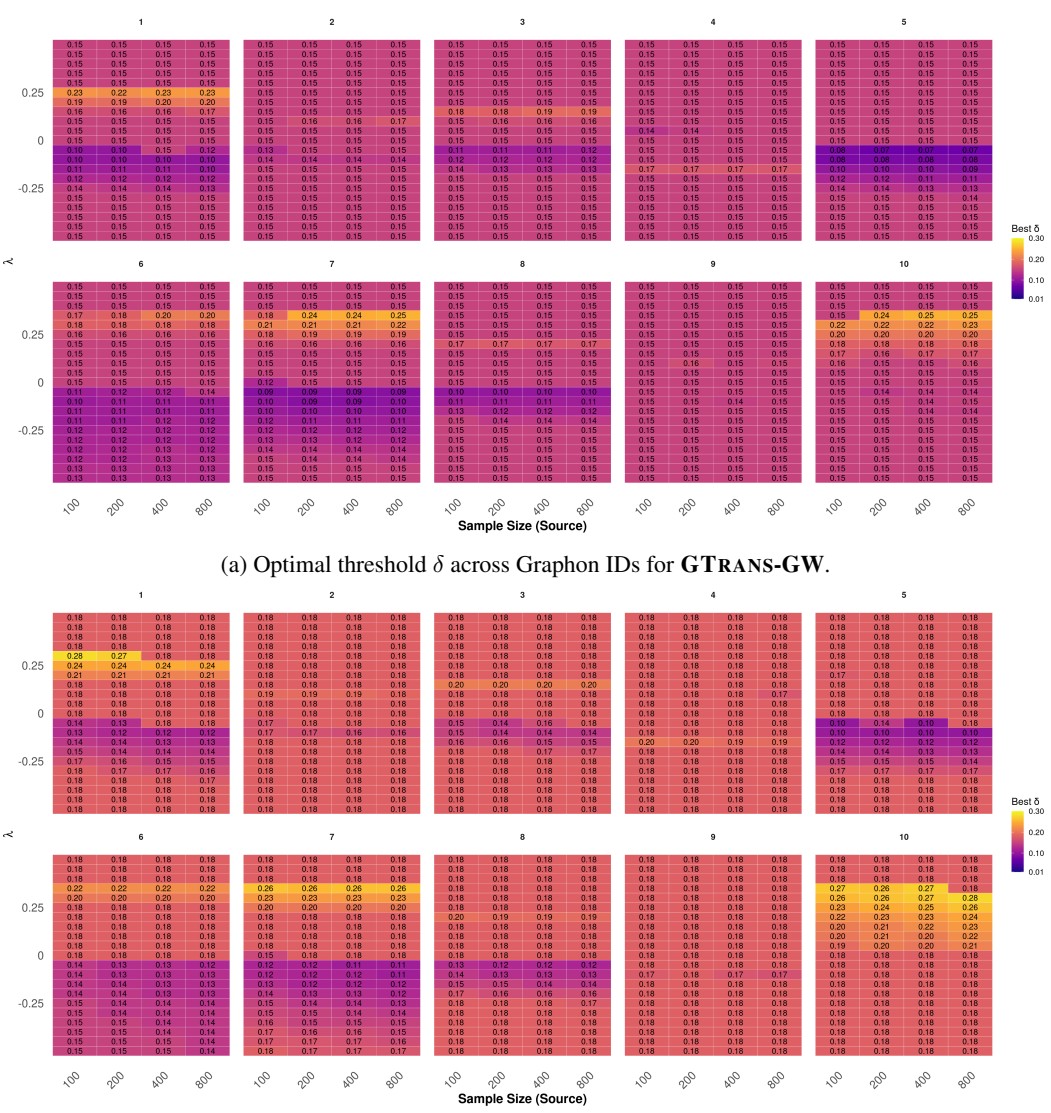

(a) Optimal threshold $\delta$ across Graphon IDs for **GTRANS-GW**.

(b) Optimal threshold $\delta$ across Graphon IDs for **GTRANS-EGW**.

Figure E3: Heatmaps of the optimal threshold $\delta$ across different Graphon IDs. **GTRANS-GW** (top) shows that $\delta = 0.15$ is consistently optimal or near-optimal, while **GTRANS-EGW** ($\epsilon = 0.01$ (bottom) favors $\delta = 0.18$ in most scenarios, with slight increases under high perturbation.

Figure E3 shows that the optimal threshold $\delta$ depends on the graphon's inherent density and the perturbation level $\lambda$. For dense graphons, $\delta$ remains stable, reflecting robustness to structural shifts. In contrast, sparse graphons exhibit a sharp increase in $\delta$ under positive $\lambda$, as perturbations enhance connectivity and reduce noise sensitivity. The asymmetry arises from a bias–variance trade-off. Denser sources ($\lambda > 0$) yield high-quality but mismatched transfer, favoring delayed debiasing with larger $\delta^*$; sparser sources ($\lambda < 0$) require earlier correction, hence smaller $\delta^*$. Besides, GW distance increases with denser sources due to structural mismatch, and decreases with sparser ones despite degraded transfer—supporting a larger $\delta$ for dense, and smaller for sparse sources.

### E.4.2 Regularization Parameter $\epsilon$ Selection

**Threshold $\epsilon$.** To determine the optimal regularization parameter $\epsilon$ in EGW, we perform $K$-fold cross-validation over a candidate set $\epsilon_{\text{list}} = \{0.001, 0.005, 0.01, 0.05, 0.1\}$. This process evaluates the MSE between the estimated and true target graphon. We observe that $\epsilon = 0.01$ is consistently selected as the best-performing choice across different settings. To further illustrate this, figure E4 presents boxplots of the average MSE for each $\epsilon$ under varying source sizes $100, 200, 300, \ldots, 1000$ with a fixed target size of $n_t = 50$. Each boxplot shows the MSE distribution across ten graphon types. While some graphons like Graphon 9 benefit from larger $\epsilon$ (e.g., 0.1) due to oscillatory patterns, $\epsilon = 0.01$ consistently achieves the lowest error, demonstrating its robustness across diverse structures.

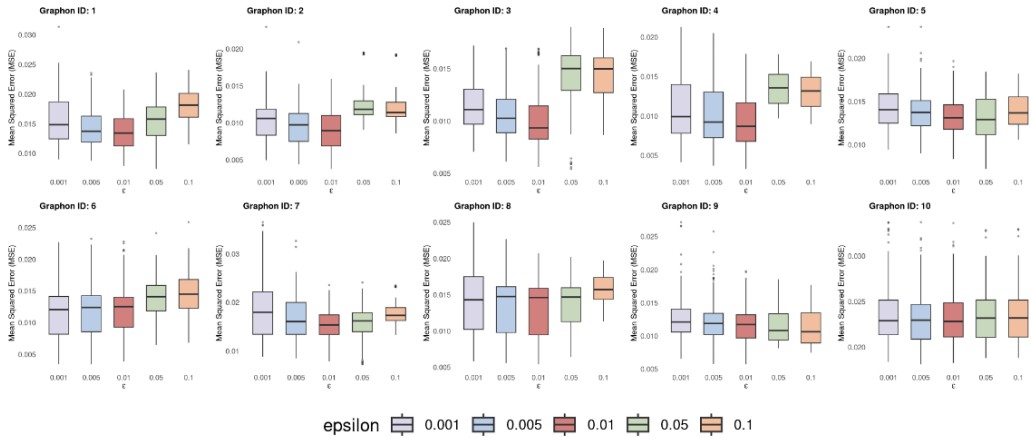

Figure E4: Boxplot comparison of different regularization parameters $\epsilon$ for GTRANS-EGW across varying source sample sizes. The results show that $\epsilon = 0.01$ (red) consistently maintains lower variance and median MSE in most graphons, indicating its robustness.

## F  Additional Real Data Results

### F.1  Additional results for graph augmentation task

In Transfer Learning, selecting the optimal source dataset and identifying the best class correspondence are crucial for effective knowledge transfer, as the structural characteristics of source labels significantly impact performance. To determine optimal class-to-class transfer pairings, we first estimate graphons separately for each source and target class using neighborhood smoothing [65]. Subsequently, we compute the pairwise Gromov–Wasserstein (GW) distances as well as the Entropic Gromov–Wasserstein (EGW) distances with a commonly chosen regularization parameter $\epsilon = 0.01$ for each target-source graphon pair, identifying the best-matched class from the source domain for each target class based on structural similarity. We summarize the key structural statistics of all datasets used in our experiments in Table F3. Table F4 summarizes the GW distances calculated between each pair of source and target labels across the considered datasets, while Table F5 presents the corresponding EGW distances, reflecting the regularized alignment between graphon pairs.

For the IMDB-Binary dataset, when transferring from Reddit-Binary, the best-matching source label for target label 1 is label 0 (GW = 0.3276, EGW = 0.3152), and for target label 2, it is also source label 0 (GW = 0.4103, EGW = 0.3987). When COLLAB is used as the source, target labels 1 and 2 best match source label 1 (GW = 0.1910, EGW = 0.1784, and GW = 0.2012, EGW = 0.1895, respectively). For IMDB-Multi, transferring from Reddit-Binary, all three target labels (1, 2, and 3) have their lowest GW distances with source label 0 (GW = 0.4437, 0.4260, and 0.5172, respectively), while the EGW distances are slightly improved (EGW = 0.4321, 0.4155, and 0.5053). When transferring from COLLAB, the best-matching source labels are more varied: target labels 1, 2, and 3 correspond best with source label 1 (GW = 0.1751, 0.2084, and 0.1552, respectively; EGW = 0.1690, 0.1968, and 0.1443). Finally, in the bioinformatics setting of PROTEINS-Full transferring from D&D, target

labels 1 and 2 both best align structurally with source label 1 (GW = 0.0616, EGW = 0.0592, and GW = 0.0652, EGW = 0.0624, respectively).

These results suggest clear structural correspondences between target and source labels, enabling effective and informed transfer learning across domains.

Table F3: Statistics of the datasets used in our experiments.

| Property | REDDIT-B | IMDB-B | IMDB-M | COLLAB | PROTEINS-FULL | D&D |
|---|---|---|---|---|---|---|
| #Graphs | 2,000 | 1,000 | 1,500 | 5,000 | 1,113 | 1,178 |
| #Classes | 2 | 2 | 3 | 3 | 2 | 2 |
| Avg. #Nodes | 429.63 | 19.77 | 13.00 | 74.49 | 25.22 | 284.32 |
| Avg. #Edges | 497.75 | 96.53 | 65.94 | 2457.78 | 226.41 | 715.66 |

Table F4: Gromov-Wasserstein distance between each source and target label pair across datasets. Bold values denote the best-matching source label for each target label.

| Source → Target | IMDB-B from REDDIT-B | | IMDB-B from COLLAB | | IMDB-M from REDDIT-B | | | IMDB-M from COLLAB | | | PROTEINS-FULL from D&D | |
|---|---|---|---|---|---|---|---|---|---|---|---|---|
| Target label | 1 | 2 | 1 | 2 | 1 | 2 | 3 | 1 | 2 | 3 | 1 | 2 |
| Source label 0 | **0.3276** | **0.4103** | 0.4710 | 0.3996 | **0.4437** | **0.4260** | **0.5172** | 0.3521 | 0.3844 | 0.3014 | 0.0631 | 0.0622 |
| Source label 1 | 0.3300 | 0.4143 | **0.1910** | **0.2012** | 0.4542 | 0.4604 | 0.5326 | **0.1751** | **0.2084** | **0.1552** | **0.0616** | **0.0652** |
| Source label 2 | – | – | 0.5475 | 0.4702 | – | – | – | 0.4092 | 0.4155 | 0.3573 | – | – |

Table F5: Entropic- Gromov-Wasserstein distance with $\epsilon = 0.01$ between each source and target label pair across datasets. Bold values denote the best-matching source label for each target label.

| Source → Target | IMDB-B from REDDIT-B | | IMDB-B from COLLAB | | IMDB-M from REDDIT-B | | | IMDB-M from COLLAB | | | PROTEINS-FULL from D&D | |
|---|---|---|---|---|---|---|---|---|---|---|---|---|
| Target label | 0 | 1 | 0 | 1 | 0 | 1 | 2 | 0 | 1 | 2 | 0 | 1 |
| Source label 0 | **0.3276** | **0.4104** | 0.4711 | 0.3996 | **0.4438** | **0.4260** | **0.5172** | 0.3523 | 0.3845 | 0.3014 | **0.0631** | 0.0584 |
| Source label 1 | 0.3300 | 0.4142 | **0.1910** | **0.1306** | 0.4549 | 0.4360 | 0.5280 | **0.1750** | **0.2085** | **0.1552** | 0.0652 | **0.0497** |
| Source label 2 | – | – | 0.5479 | 0.4773 | – | – | – | 0.4092 | 0.4155 | 0.3573 | – | – |

We also include two transfer baselines. (1) **Pooled Estimation** – pooling source & target graphs and estimating a graphon using the G-Mixup procedure with neighborhood smoothing (NS), and (2) **Pooled-then-Transfer** – using the pooled estimator as the initial source estimate $\hat{P}_s^{\text{ini}}$ in our transfer framework, followed by the GTRANS procedure (denoted PGTRANS-GW & PGTRANS-EGW). From Table F6, both pooled baselines underperform our method, suggesting that naive joint estimation is suboptimal when source and target are misaligned, while our alignment and debiasing steps enable more accurate transfer.

Table F6: Graph classification accuracy (%) comparison between GTRANS and pooled transfer estimation methods (mean ± std).

| Source | Target | GTRANS-GW | GTRANS-EGW | Pooled NS | PGTRANS-GW | PGTRANS-EGW |
|---|---|---|---|---|---|---|
| Reddit-B | IMDB-B | 76.30 ± 2.35 | **76.80 ± 1.52** | 74.30 ± 2.52 | 74.65 ± 2.64 | 75.15 ± 3.03 |
| COLLAB | IMDB-B | 76.25 ± 2.06 | **77.50 ± 2.13** | 74.15 ± 2.26 | 73.90 ± 2.34 | 74.75 ± 1.89 |
| Reddit-B | IMDB-M | 49.10 ± 1.33 | **51.27 ± 1.98** | 46.50 ± 3.13 | 48.10 ± 3.30 | 48.67 ± 3.04 |
| COLLAB | IMDB-M | **50.47 ± 1.42** | 50.23 ± 0.92 | 47.70 ± 2.86 | 49.60 ± 2.85 | 48.97 ± 1.79 |
| D&D | PROTEINS | **69.33 ± 2.55** | 68.52 ± 1.59 | 66.59 ± 2.19 | 67.80 ± 2.73 | 67.53 ± 2.37 |

## F.2 Application to Link Prediction

Existing literature applied graphon estimation to the link prediction task [65, 45], where the goal is to predict missing or future connections between nodes in a network.

### F.2.1 Transfer from a Single Source Network

**Datasets.** We evaluate our method on five real-world social networks spanning diverse domains: dolphins, karate, football, firm, and wiki-vote. These datasets cover different interaction types, including animal social behavior (dolphins), human relationships and organizational affiliations (karate, firm), and communication or membership networks (football, wiki-vote). Table F7 summarizes their key statistics. The number of nodes ranges from 33 (firm) to 2,375 (wiki-vote), and the number of edges from 78 (karate) to 16,717 (wiki-vote). The average degree varies accordingly, from 4.59 (karate) to 14.07 (wiki-vote). Network density also differs widely, with firm being relatively dense (0.1723) while wiki-vote is much sparser (0.0074). We select wiki-vote as the source network because it is the largest. Link prediction is then performed on each of the remaining datasets treated as targets, with results averaged over 50 random seeds.

Table F7: Basic statistics of the real-world datasets used in graphon estimation.

| Dataset | #Nodes | #Edges | Avg. Deg. | Density |
|---------|--------|--------|-----------|---------|
| dolphins | 62 | 159 | 5.13 | 0.0841 |
| karate | 34 | 78 | 4.59 | 0.1390 |
| football | 115 | 613 | 10.66 | 0.0935 |
| firm | 33 | 91 | 5.52 | 0.1723 |
| wiki-vote | 889 | 2914 | 6.56 | 0.0074 |

**Experimental Setup.** We simulate a realistic link prediction task using a masking-based evaluation strategy following [65]. Specifically, we randomly mask a subset of edges in the upper triangular portion of the target adjacency matrix to form a test set. Let $M \in \{0,1\}^{n \times n}$ be a masking matrix with $M_{ij} \sim \text{Bernoulli}(1 - p)$, where $p$ is the test ratio. The observed matrix is $\mathbf{A}_{ij}^{\text{mask}} = M_{ij}\mathbf{A}_{t,ij}$, meaning each edge is observed with probability $1 - p$. We set $p = 0.1$ unless otherwise specified. Both **GTRANS-GW** and **GTRANS-EGW** are applied to $\mathbf{A}^{\text{mask}}$ to estimate $\hat{\mathbf{P}}_t$, and each experiment is repeated 50 times with different seeds to report averaged performance.

**Evaluation.** To evaluate the performance of link prediction, we computed the area under the receiver operating characteristic curve (AUC). Evaluation is based on the masked (unobserved) entries where $M_{ij} = 0$, using the original adjacency $\mathbf{A}_t$ as ground truth. For a threshold $t > 0$, the false positive rate (FPR) and true positive rate (TPR) are defined as:

$$r_{\text{FP}}(t) = \frac{\sum_{ij} \mathbf{1}\left(\hat{\mathbf{P}}_{t,ij} > t,\ A_{ij}^{\text{true}} = 0,\ M_{ij} = 0\right)}{\sum_{ij} \mathbf{1}\left(\mathbf{A}_{t,ij} = 0,\ M_{ij} = 0\right)}$$

$$r_{\text{TP}}(t) = \frac{\sum_{ij} \mathbf{1}\left(\hat{\mathbf{P}}_{t,ij} > t,\ \mathbf{A}_{t,ij} = 1,\ M_{ij} = 0\right)}{\sum_{ij} \mathbf{1}\left(\mathbf{A}_{t,ij} = 1,\ M_{ij} = 0\right)}$$

These quantities are used to construct the ROC curve by varying the threshold $t$, and the AUC is computed as the area under this curve.

**Results.** We evaluate transfer performance using Wiki-Vote as the source network ($n_s = 889$). Each of the remaining datasets (dolphins, firm, football, karate) is treated as the target graph. Results are averaged over 50 random seeds. Table F8 reports the link prediction AUC (mean $\pm$ standard deviation, multiplied by 100). Our method consistently matches or outperforms baselines across datasets. Overall, these results highlight the effectiveness and robustness of transfer-based graphon estimation, particularly in smaller or noisier networks where leveraging external structure is beneficial.

Table F8: Link prediction AUC scores (mean $\pm$ std, $\times 100$). Best result per dataset is **bolded**.

| Dataset | GTRANS-GW | GTRANS-EGW | NS | USVT | SAS | ICE |
|---------|-----------|------------|-----|------|-----|-----|
| Dolphins | 75.96±8.53 | **76.26±8.54** | 70.60±9.01 | 72.36±10.16 | 50.66±6.35 | 72.77±9.59 |
| Firm | **71.31±12.18** | 71.26±12.06 | 66.32±12.27 | 65.56±12.25 | 54.90±7.59 | 67.60±12.24 |
| Football | 86.64±3.66 | 86.74±3.72 | **86.75±3.49** | 85.32±3.56 | 44.56±7.38 | 81.83±4.60 |
| Karate | 82.47±10.36 | **82.53±10.46** | 76.74±12.01 | 71.86±15.20 | 63.88±11.15 | 77.20±10.82 |

### F.2.2 Intra- vs. Inter-Dataset Transfer

**Datasets.** We evaluate intra- and inter-dataset transfer on IMDB-BINARY. For each target dataset, the smallest graph ($> 20$ edges) is used as the target. As sources, we consider (i) the largest graph from the same dataset, (ii) the structurally closest graph within the same dataset (smallest GW distance), and (iii) the largest graphs from Reddit-BINARY or COLLAB.

**Experimental Setup.** Graphons are estimated using GTRANS-GW and compared against NS, ICE, SAS, and USVT.

**Results.** Table F9 reports AUC scores. Transfer from the most structurally similar IMDB-B graph achieves the best performance (AUC 0.96), surpassing both the largest IMDB-B source and all inter-dataset sources. This highlights that transfer is most effective when the source and target share high structural similarity (small GW distance).

Table F9: Link prediction AUC results on target graphs with various transfer sources. For brevity, standard deviations are omitted.

| Source | Target | $n_s$ | $n_t$ | GW | GTRANS-GW | NS | ICE | SAS | USVT |
|---|---|---|---|---|---|---|---|---|---|
| IMDB-B (largest) | IMDB-B | 136 | 12 | 0.39 | 0.94 | 0.91 | 0.79 | 0.59 | 0.75 |
| IMDB-B (GW smaller) | IMDB-B | 84 | 12 | 0.31 | **0.96** | 0.91 | 0.79 | 0.59 | 0.75 |
| Reddit-B | IMDB-B | 41 | 12 | 0.40 | 0.94 | 0.91 | 0.79 | 0.59 | 0.75 |
| COLLAB | IMDB-B | 191 | 12 | 0.43 | 0.94 | 0.91 | 0.79 | 0.59 | 0.75 |
| IMDB-M | IMDB-B | 89 | 12 | 0.40 | 0.95 | 0.91 | 0.79 | 0.59 | 0.75 |

## G  Additional Details

### G.1  Graphon functions

We implement 10 distinct graphon structures (see Table G10) ranging from simple bilinear forms to highly structured oscillatory and piecewise functions.

Table G10: Graphon functions implemented.

| ID | Graphon Function |
|---|---|
| 1 | $\exp(-x^{0.7} - y^{0.7})$ |
| 2 | $\exp(-\max(x, y)^{0.75})$ |
| 3 | $\exp\left(-0.5\left[\min(x, y) + \sqrt{x} + \sqrt{y}\right]\right)$ |
| 4 | $\frac{1}{1+\exp(-[\max(x,y)^2+\min(x,y)^4])}$ |
| 5 | $|x - y|$ |
| 6 | $\frac{xy}{2}$ |
| 7 | $\frac{x^2+y^2}{3}\cos\left(\frac{1}{x^2+y^2}\right) + 0.15$ |
| 8 | $\frac{x+y}{3}\cos\left(\frac{1}{x+y}\right) + 0.15$ |
| 9 | $\frac{\sin(10\pi(x+y-5))}{5} + 0.5$ |
| 10 | $\frac{1}{4}\min\left(\exp\left(\sin\left(\frac{6}{(1-x)^2+y^2}\right)\right), \exp\left(\sin\left(\frac{6}{x^2+(1-y)^2}\right)\right)\right)$ |

### G.2  Neighborhood Smoothing Details

For effective neighborhood smoothing, we must identify a neighborhood $\mathcal{N}_i$ for each node $i$ that contains only nodes with approximately homogeneous distribution. Here, the neighborhood $\mathcal{N}_i$ in this context refers to nodes that are *close in the underlying latent space* (i.e., have similar latent positions), not nodes that are connected in the observed graph. Mathematically, node $i'$ belongs to node $i$'s neighborhood if their graphon values are sufficiently close: $\|f(u_i, \cdot) - f(u_{i'}, \cdot)\|_2 \leq \eta$, where $\|\cdot\|_2$ denotes the $L_2$ norm with inner product $< f, g >= \int_0^1 f(x)g(x)dx$, and $\eta$ is a tolerance parameter. Empirically, this can be estimated by $\sum_{j=1}^n \|\hat{p}_{ij} - \hat{p}_{i'j}\|_2^2$. As proved in [65], under mild conditions,

this quantity is bounded by: $\Delta_{ij}^2 = \max_{k \neq i,i'} \frac{1}{n} |\sum_{j=1}^{n} (\mathbf{A}_{ij} - \mathbf{A}_{i'j})\mathbf{A}_{kj}|$. This bound enables efficient neighborhood definition: $\mathcal{N}_i = \{i' \neq i : \Delta_{ii'} \leq \tau\}$, where $\tau$ is a predefined threshold. Once neighborhoods $\mathcal{N}_i$ and $\mathcal{N}_j$ are identified, nodes within each neighborhood are treated as replicates.

Thus, $\hat{\mathbf{P}}_{ij}$ can be estimated by neighbourhood smoothing, i.e., $\hat{\mathbf{P}}_{ij} = \frac{1}{2}(\frac{\sum_{i' \in \mathcal{N}_i} \mathbf{A}_{i'j}}{|\mathcal{N}_i|} + \frac{\sum_{j' \in \mathcal{N}_j} \mathbf{A}_{ij'}}{|\mathcal{N}_j|})$, where $|\mathcal{N}_i|$ represents the number of nodes in the neighborhood $\mathcal{N}_i$. This averages two empirical proportions: edges between node $j$ and nodes in $\mathcal{N}_i$, and edges between node $i$ and nodes in $\mathcal{N}_j$.

### G.3 Algorithm

#### G.3.1 Algorithm: GTRANS

---

**Algorithm 1** GTRANS: Transfer Learning for Graphon Estimation

---

**Require:** Source adjacency matrix $\mathbf{A}_s$, target adjacency matrix $\mathbf{A}_t$, a threshold $\epsilon$.
**Ensure:** Target graphon estimator $\hat{\mathbf{P}}_t$
 1: **Step 1: Initial Graphon Estimation**
 2: Apply neighborhood smoothing to obtain two initial estimators for both the source and target graph: $\hat{\mathbf{P}}_s^{ini} \leftarrow NS(A_s)$, $\hat{\mathbf{P}}_t^{ini} \leftarrow NS(A_t)$.
 3: **Step 2: Transferring Step**
 4: Compute the optimal transport plan $\hat{\pi}$ and the optimal transportation distance $d$ between $\hat{\mathbf{P}}_s^{ini}$ and $\hat{\mathbf{P}}_t^{ini}$.
 5: Apply column normalization to $\hat{\pi}$, obtaining $\tilde{\pi}$
 6: Transfer source graphon estimator to target domain: $\hat{\mathbf{P}}_t^{trans} \leftarrow \tilde{\pi}^T \hat{\mathbf{P}}_s^{ini} \tilde{\pi}$.
 7: Refine the transferred estimator by applying neighborhood smoothing to $\hat{\mathbf{P}}_t^{trans}$: $\hat{\mathbf{P}}_t^{trans2} \leftarrow NS(\hat{\mathbf{P}}_t^{trans})$.
 8: **if** $d > \delta$, **then**
 9:     **Step 3: Debiasing Step:**
10:     Compute residual matrix: $\mathbf{R}_t \leftarrow \hat{\mathbf{P}}_t^{ini} - \hat{\mathbf{P}}_t^{trans2}$
11:     Apply neighborhood smoothing to residual, obtaining estimator $\hat{\mathbf{P}}_t^{res}$.
12:     Combine transferred estimator with smoothed residual: $\hat{\mathbf{P}}_t \leftarrow \hat{\mathbf{P}}_t^{trans2} + \hat{\mathbf{P}}_t^{res}$
13: **else**
14:     $\hat{\mathbf{P}}_t \leftarrow \hat{\mathbf{P}}_t^{trans2}$
15: **end if**
16: **return** $\hat{\mathbf{P}}_t$

---

#### G.3.2 Algorithm: Network Cross-Validation by Edge Sampling

---

**Algorithm 2** Network Cross-Validation by Edge Sampling for Threshold Selection

---

**Require:** Adjacency matrices $\mathbf{A}_s, \mathbf{A}_t$, true probability matrix $\mathbf{P}_t$, threshold candidates $\{\epsilon_1, \ldots, \epsilon_L\}$, number of folds $K$, loss function $\mathcal{L}(\cdot, \cdot)$
**Ensure:** Selected threshold $\hat{\epsilon}$, final loss, estimated graphon $\hat{\mathbf{P}}_t$
 1: **Step 1: Edge Sampling**
    Randomly partition the edges of $\mathbf{A}_t$ into $K$ folds, ensuring that each fold is disjoint and collectively covers the entire set of edges.
 2: **for** $k = 1$ to $K$ **do**
 3:     Mask edges in the $k$-th fold to form the incomplete adjacency matrix $\mathbf{A}_t^{(k)}$.
 4:     Perform matrix completion as suggested by [33], leveraging low-rank and smoothness assumptions to recover the completed adjacency matrix.
 5:     Apply GTRANS with each threshold $\epsilon_l$ and evaluate the loss $\mathcal{L}^{(k)}(\epsilon_l)$ on the held-out set $\Omega_c^{(k)}$
 6: **end for**
 7: **Step 2: Select Optimal Threshold**
 8: Average the loss across all folds: $\bar{\mathcal{L}}(\epsilon_l) = \frac{1}{K} \sum_{k=1}^{K} \mathcal{L}^{(k)}(\epsilon_l)$
 9: Select the threshold that minimizes the average loss: $\hat{\epsilon} = \arg\min_{\epsilon_l} \bar{\mathcal{L}}(\epsilon_l)$
10: **Return** $\hat{\epsilon}$.

---

### G.4 Graph Augmentation Model Training Setup.

We follow a modified version of the training configuration from [22]. Specifically, we train a GIN model for 200 epochs using the Adam optimizer with a fixed learning rate of 0.01. The mini-batch size is set to 128, and the hidden dimension is 64. Validation loss is monitored throughout training, and test accuracy is reported at the epoch with the best validation performance.

