# OpenReview forum: "Transfer Learning on Edge Connecting Probability Estimation Under Graphon Model"
_NeurIPS.cc/2025/Conference — NeurIPS 2025 poster_

### Official Review · Reviewer_81ne · 2025-07-02

**Clarity:** 3
**Significance:** 3
**Originality:** 3
**Rating:** 5
**Confidence:** 4

**Summary:**

This paper proposes to use Gromov-Wasserstein optimal transport to improve graphon estimation in small graphs. The paper gives theoretical guarantees and illustrations using experiments.

**Questions:**

Please clarify whether the graph is assumed to have selfloops or not.
Why is the MSE divided by n^2 when the graphon is symmetric?
Is a GW optimal coupling unique? Otherwise change `the' to `a', and specify which one you pick.]
What are \nu and \mu in your formulation, and what is their product?
How do you obtain initial estimates for the GTrans method?
in the graphon model, the edge indicators are dependent unless the U_i's are given. How is this dependence reflected in the method?
Theorem 4.1 is phrased in terms of a cutoff (which would be interesting to mention in the main text. Could you also take randomness into account and obtain a `with high probability' statement?
Asymptotic performance: what is the symbol combination after source data?
There is a vast literature on graph classification, including non-parametric methods based on graphlets. It could be interesting to see whether your method improes on these.
The method works only for small graphs, but that is clearly explained. It could be interesting to get a quantitative bound in terms of the graph size, or some indication of what counts as small in the paper.

**Ethical Concerns:**

["NO or VERY MINOR ethics concerns only"]

**Final Justification:**

The authors have addressed my concerns. In my view this is a good paper which should be accepted.

**Limitations:**

As the  work is theoretical, the fact that  potential societal impact is not discussed is not an issue..

**Paper Formatting Concerns:**

Figure 2: the headers are too small for me to read.

**Quality:**

3

**Strengths And Weaknesses:**

The approach in the paper makes sense and is theoretically founded. Some presentation issues should be addresses, see the questions below.

The estimating method assumes that the graph is a small subgraph of a larger source graph with related structural similarities. In particular it is assumed that the initial estimates of the edge probabilities, obtained from the source graph, are informative. This assumption limits the applicability of the method.

It is a strength of the paper that it makes this assumption clear.It is a standard assumption in transfer learning.

Using a Gromov-Wasserstein distance makes theoretical sense but it is computationally intensive. The use of an entropic regulariser alleviates the computational complexity, However the theoretical results seem to be derived for the unregularised version (although it is not clearly stated in Theorem 4.1 what the EGW optimization problem is). The assumptions for the theoretical guarantees may not be easy to check as the true distribution is unknown.

---

> ### Author Rebuttal · Authors · 2025-07-31
>
> We thank the reviewer for these constructive comments. We will improve figure quality, and add the following clarifications and discussions in the revision.
>
> >Comment 1: Assumption on similarity between source and target
>
> Response: We agree that the effectiveness of GTRANS depends on the assumption that the target graph exhibits structural similarity to a larger source graph, as is standard in transfer learning.
>
> However, we would like to highlight the fact that our method is robust even in the absence of strong source-target similarity. When GW distance is large, our debiasing step ensures GTRANS performs on par with target-only NS, as shown in Table 1 for dissimilar pairs like 8 and 9. This robustness means that, **while structural similarity enhances transfer effectiveness, GTRANS remains a safe and broadly applicable tool even when ideal source-target matching is not available**. We will further clarify in the revised manuscript.
>
> >Comment 2: Theorem 4.1 is for Entropic GW or GW?
>
> Response:   Theorem 4.1 is for the Entropic Gromov-Wasserstein (EGW) distance instead of GW distance. The $\epsilon$ in the statement refers to the entropic regularization parameter. We will state it more clearly in the revised version of the manuscript.
>
> >Comment 3: Validating assumptions of Theorem 4.1
>
> Response:  Theorem 4.1 relies on two key conditions: (i) $\epsilon > C_1|\pi^\star|_\infty |\mathbf{P}_s \otimes \mathbf{P}_t|{\rm op}$ for some $C_1 > 2$, and (ii) the estimation errors for $\mathbf{P}_s$ and $\mathbf{P}_t$ must be sufficiently small.
>
> While the **first condition** involves unobservable population-level quantities and cannot be directly verified, our simulations suggest it holds for a broad range of $\epsilon$. **Table E** (please find in our response to Reviewer aKUK) reports computed lower bounds on feasible $\epsilon$ values (with $C_1 = 2.01$) for Graphons 1–3 (defined in Section E.1), each scaled to have operator norm 1, with $n_s = 100$ and $n_t = 50$. Any $\epsilon$ above the reported thresholds satisfies the condition, including $\epsilon = 0.01$ used in our real data analysis.
>
> The second condition is well-supported by established graphon estimation literature, as it requires $\hat{\mathbf{P}}_s^{\text{ini}}$ and $\hat{\mathbf{P}}_t^{\text{ini}}$ to be consistent. For example, Ref. [62] shows that NS-based estimators achieve rates of $\sqrt{\log n_s / n_s}$ and $\sqrt{\log n_t / n_t}$ for the source and target, respectively.
>
> >Comment 4. **Whether selfloops or not, Why is the MSE divided by n^2**:
>
> Response: **Self-loops:** Our theoretical results can be applied to both networks with and without self-loops. In our simulation experiments, for simplicity, we conducted exepriments on networks without self-loops. Our real-data also does not contain self-loop.
>
> **MSE Normalization:** The MSE is calculated as the average squared error over all $n^2$ entries in the probability matrix, following conventions established in prior graphon estimation studies [1-3]. While the graphon is symmetric, this normalization is widely adopted for consistency and comparability across the literature.
>
> [1] Zhang, Y., Levina, E., & Zhu, J. (2017). Estimating network edge probabilities by neighbourhood smoothing. Biometrika, 104(4), 771-783.
>
> [2] Chan, S., & Airoldi, E. (2014). A consistent histogram estimator for exchangeable graph models. In International Conference on Machine Learning (pp. 208-216). PMLR.
>
> [3] Qin, Y., Yu, L., & Li, Y. (2021). Iterative connecting probability estimation for networks. Advances in Neural Information Processing Systems, 34, 1155-1166.
>
>
> >Comment 5. **GW optimal coupling uniqueness**:
>
> Response:  In general, the Gromov-Wasserstein (GW) optimization problem is **non-convex**, so multiple optimal or near-optimal coupling matrices may exist in theory. To clarify our implementation, we use the `ot.gromov_wasserstein` function from the POT library [1] in Python to compute the GW distance and corresponding coupling. While the GW problem may have multiple optima, the POT algorithm deterministically converges to a single stationary point for fixed inputs and initialization. This ensures reproducibility, though not mathematical uniqueness. We will revise the manuscript to state "an optimal coupling" and clarify.
>
> [1] Flamary, R., Courty, N., Gramfort, A., et al. (2021). POT: Python Optimal Transport. *Journal of Machine Learning Research*, 22(78), 1–8.
>
> >Comment 6. **\nu and \mu**:
>
> Response:  In our formulation, $\mu$ and $\nu$ denote the uniform measures on the source and target nodes, respectively; that is, $\mu = (\underbrace{n\_S^{-1}, \dots, n\_S^{-1}}\_{:= n\_S \text{ times}})$ and $\nu = (\underbrace{n\_T^{-1}, \dots, n\_T^{-1}}\_{:= n\_T \text{ times}})$. Their product, $\mu \otimes \nu$, is the product measure, which assigns a uniform weight of $(n_S n_T)^{-1}$ to each possible pair consisting of a source and a target node.
>
> >Comment 7. **How to obtain initial estimates?**:
>
> Response:  **Initial Estimates:** We obtain the initial edge probability matrices  using the **neighborhood smoothing (NS)** method  [1] (see lines 60–63 in our paper).
>
> **Dependence in the Graphon Model:** You are correct that, in the graphon model, edge indicators such as $A_{ij}$ and $A_{ik}$ are dependent because they share the same latent variable $u_i$. Rather than ignoring or explicitly modeling these dependencies, NS actually **leverages** them. For each node $i$, NS defines a neighborhood $\mathcal{N}_i$ consisting of nodes $j$ whose latent positions $u_j$ are close to $u_i$, which means their edge probabilities pattern are similar. Since the latent variables $u_j$ are unobserved, in practice, NS approximates $\mathcal{N}_i$ by identifying nodes $j$ whose observed connectivity patterns (i.e., rows of the adjacency matrix $A$) are close to that of node $i$ under the $L_2$ norm. NS averages edge values within empirically defined neighborhoods, pooling information from correlated edges to consistently estimate edge probabilities—much like kernel smoothing in nonparametric regression.
>
> [1] Zhang, Y., Levina, E., & Zhu, J. (2017). Estimating network edge probabilities by neighborhood smoothing. *Biometrika*, 104(4), 771–783.
>
> >Comment 8. **Theorem 4.1: Consider adding the cutoff and strengthening it to a high-probability statement**:
>
> Response: Indeed as the estimation error $||\hat{\mathbf{P}}\_s^{ini} - \mathbf{P}\_s||\_\infty$ and $||\hat{\mathbf{P}}\_t^{ini} - \mathbf{P}\_t||\_\infty$ decays at the order $\sqrt{\log{n\_s}/n\_s}$ and $\sqrt{\log{n\_t}/n\_t}$ respectively under certain assumptions on the underlying graphon (e.g., see Theorem 2 of \cite{}), we can leverage this type of result to argue that the conclusion of our theorem holds with probablity $\ge 1 - C(n\_s \wedge n\_t)^{-c}$ for some constant $C , c > 0$. We will update the description of the theorem in the revised version of the manuscript as per your suggestion.
>
> >Comment 9. **symbol after "source data."**
>
> Response:  We are not entirely certain what is meant by “symbol combination after source data.” We assume you are referring to the Kronecker product notation (e.g., $\mathbf{P}_s \otimes \mathbf{P}_t$) that appears in Theorem 4.1. In particular, the Kronecker product between two matrices is a way of combining them to create a larger matrix. This operation expands each entry of the first matrix by multiplying it with the entire second matrix, capturing all pairwise interactions between elements—such as nodes in source and target graphs. In our context, it enables comprehensive comparison within the Gromov-Wasserstein framework. If this doesn't fully address your question, we’d appreciate further clarification.
>
> >Comment 10. **Comparing graphlet-based methods on graph classifcation**:
>
> Response:  In response, we have included a comparison between our method and a representative graphlet-based graph classification approach. Specifically, we implemented the Graphlet Kernel method.  We use the Graphlet Kernel from the GraKeL library in Python to compute pairwise graph similarities, followed by SVM classification.  As shown in Table R6, our transfer-based graphon method significantly outperforms Graphlet on  two target data (IMDB-BINARY and IMDB-MULTI), while achieving comparable performance as Graphlet on PROTEINS target data.
>
> Table R6: Graph classification accuracy (\%) across three target datasets
> | Source   | Target   | GTRANS-GW        | GTRANS-EGW       | Graphlet          |
> | -------- | -------- | ---------------- | ---------------- | ------------
> | Reddit-B | IMDB-B   | 76.30     | **76.80** | 61.10  |
> | COLLAB   | IMDB-B   | 76.25  | **77.50** | 61.10 |
> | Reddit-B | IMDB-M   | 49.10     | **51.27** | 39.37  |
> | COLLAB   | IMDB-M   | **50.47** | 50.23    | 39.37  |
> | D\&D     | PROTEINS | **69.33** | 68.52     | 70.11
>
> >Comment 11: **quantitative bound for how “small” graph means**:
>
> Response: Defining universal bounds on "small" $n$ is challenging as it depends on graphon complexity and source-target similarity rather than fixed thresholds. Complex graphons require larger $n_t$. Existing works provide asymptotic error rates (e.g., $O(\sqrt{\log n / n})$ for NS method) but lack specific "small" cutoffs.
>
> To provide practical guidance, we ran simulations with Graphon 1, fixing $n_s = 1000$ and varying $n_t$ from 100 to 700 (**Table R4**, in our response to Reviewer JN9a).  The relative gain from transfer learning decreases as $n_t$ increases but remains positive; even at $n_t = 700$, GTRANS still outperforms target-only NS, showing that transfer learning provides substantial benefit even for moderately large target graphs.

---

> > ### Comment · Reviewer_81ne · 2025-08-01
> >
> > Thank you for the clarification. I have no further questions and I shall not change my score.

---

> > > ### Author Response · Authors · 2025-08-01
> > > **Thank you**
> > >
> > > We are glad our response has addressed your concern. Thank you again for these constructive comments, which help us improve the manuscript.

---

### Official Review · Reviewer_JN9a · 2025-07-02

**Clarity:** 4
**Significance:** 4
**Originality:** 3
**Rating:** 5
**Confidence:** 3

**Summary:**

This paper studies the problem of graphon matrix estimation via transfer learning, where the goal is to estimate the target probability matrix by additionally leveraging information from a larger source graph. The paper establishes an adaptive transfer learning framework under the graphon model  by combining neighborhood smoothing graphon estimation with Gromov-Wasserstein optimal transport which maps the source probability matrix estimate to the target  probability matrix estimate. When the discrepancy between the target and the transferred source estimate is larger than a pre-specified threshold, a smoothed version of the discrepancy matrix is used to improve the proposed estimator (referred to as 'de-biasing' in the paper).

**Questions:**

Major:
1. Why is the debiasing step referred to as debiasing? What is the bias here and how is it being removed?
2. What makes the debiasing step adaptive?
3. Line 41: please explicitly define `the source-target domain shift'
4. Why is the difference between \hat P_t^ini  and \hat P_t^trans2 called a `residual matrix' and not simply a discrepancy matrix? The term `residual' is commonly used in the context of linear regression where it represents the structure unexplained by the observed predictors. Is there a link/analogy that can be made?
5. Lines 189-190: why is this additional smoothing required? Does the transfer step lead to discontinuities in the smoothed \hat P^ini_s? Can this be theoretically justified or illustrated empirically?
6. Line 229- where is the lower bound on \epsilon?
7. Theorem 4.1: Does this cover the specific case where the source graph is completely uninformative about the target graph? What would happen in that case?
8. Experiments: The authors study the performance with n_t=50 and n_s \in { 100,200,..1000}. It would be helpful to understand how the method compares to the baseline NS method when n_t =100, n_s=150, 200 and when n_t=150 and n_s=200, 250. This could serve as a practical guidance on when transfer learning may or may not be useful.
9. General guidance on properties to look for when selecting the source graph would be very useful.

Minor

10. Lines 54-57: Are the two problems (alignment and unsupervised learning) linked?
11. Section 2.1: The list of references on graphon estimation could be updated to include more recent significant contributions on the topic

**Ethical Concerns:**

["NO or VERY MINOR ethics concerns only"]

**Final Justification:**

I thank the authors for their clear response which have improved my appreciation for the paper. I have increased my scores accordingly.

**Limitations:**

Yes. However, limitations within the framework studied in the paper are missing and could be added.

**Paper Formatting Concerns:**

Typo: line 139: first 'between' can be removed

**Quality:**

3

**Strengths And Weaknesses:**

Strengths: The proposed method is shown to work in the sense that it leads to an improved graphon estimator in some sample settings (n_s,n_t) where the target graph is small and the source graph has structure to some extent similar to the target graph. Theoretical consistency of the estimated alignment matrix to the true optimal transport map is established. The paper is clearly written.

Weakness: The main motivation behind the contribution is relatively poor graphon estimates when n is small. However, it is not clear what defines a `small' n, in practice. If the underlying graphon structure is simple (e.g. an ER or a 2-blockmodel), n as small as 80 might be sufficient to estimate it. This is not discussed in the paper at all. Further, there are no theoretical results which establish sample settings (n_s,n_t) and assumptions on the underlying target and source graphons, where we may expect the proposed methodology to be useful.

---

> ### Author Rebuttal · Authors · 2025-07-31
>
> We thank the reviewer for these constructive comments. We will correct typos, add the following clarifications and discussions in the revision. Due to space constraints, only mean values are shown in the following experiments.
>
> > Comment 1: How small should the target data be?
>
> Response: Defining universal bounds on "small" $n$ is challenging as it depends on graphon complexity and source-target similarity rather than fixed thresholds. Complex graphons require larger $n_t$. Existing works provide asymptotic error rates (e.g., $O(\sqrt{\log n / n})$ for NS method) but lack specific "small" cutoffs.
>
> To provide practical guidance, we ran simulations with Graphon 1, fixing $n_s = 1000$ and varying $n_t$ from 100 to 700 (Table R4, 50 runs each).  The relative gain, calculated as $\frac{MSE_{NS} - MSE_{GTRANS}}{MSE_{NS}}$ (where $MSE_{GTRANS}$ is the minimum of GTRANS-EGW and GTRANS-GW). The relative gain from transfer decreases as $n_t$ increases but remains positive; even at $n_t = 700$, GTRANS still outperforms target-only NS, showing that transfer learning provides substantial benefit even for moderately large target graphs.
>
> Table R4. Mean MSE  value (multiplied by 100) over 50 runs for Graphon 1 with  $n_s = 1000$.
>
> | $n_t$ | GTrans-EGW  | GTrans-GW  | NS | Relative Gain (%) |
> | ----------- | -------------- | ------------- | -------------------- | -------------------- |
> | 100         | 1.34     | 1.44    | 1.83          | 26.78               |
> | 300         | 1.11     | 1.21    | 1.40           | 20.71              |
> | 500         | 1.09     | 1.19    | 1.31          | 16.79               |
> | 700         | 1.10    | 1.20   | 1.26           | 12.70               |
>
> > Comment 2:  lack of theoretical conditions on the target and source graphons under which the proposed method is effective.
>
> Response: We derived new theoretical results which can help us understand **how close the graphons should be** for effective transfer.
>
> (1) **Theoretical Justification for GW Distance as Similarity Measure**: We developed new theory showing that GW distance between initial estimators $\hat{\mathbf{P}}^{ini}_s$ and $\hat{\mathbf{P}}^{ini}_t$ is a theoretically justified similarity criterion. Our results establish: (i) $GW_2^2(\mathbf{P}_s, \mathbf{P}_t) \lesssim GW_2^2(\hat{\mathbf{P}}^{ini}_s, \hat{\mathbf{P}}^{ini}_t) + \delta_n$ and (ii) $GW_2^2(\hat{\mathbf{P}}^{ini}_s, \hat{\mathbf{P}}^{ini}_t) \lesssim GW_2^2(\mathbf{P}_s, \mathbf{P}_t) + \delta_n$, where $\delta_n$ represents estimation error of order $\sqrt{\log(n)/n}$ for Lipschitz smooth graphons [1]. **Together**, these results establish that GW distance between initial estimators reliably proxies true graphon similarity, justifying its use for guiding effective transfer learning.
>
> (2) **Theoretical Result of Transferring Step**:  Our theoretical analysis shows that at the population level, the distance between the transferred estimator and true target probability matrix is bounded by the source-target GW distance: $|{\mathbf{P}}_t^{\rm trans} - \mathbf{P}_t| \leq \mathrm{GW}_2^2(\mathbf{P}_s, \mathbf{P}_t)$. This demonstrates that the transferring step effectively transfers structural information, particularly when source and target graphons are sufficiently similar.
>
> > Comment 3: Why called  "debiasing step"?
>
> Response: When source and target probability matrices ($\mathbf{P}_s$ and $\mathbf{P}_t$) differ, the transferred estimator may systematically misestimate target probabilities due to structural differences, creating bias that requires correction. "Debiasing" is standard terminology in transfer learning literature [1].
>
> **Why Debiasing helps**:  The residual matrix $\mathbf{R}_t = \hat{\mathbf{P}}^{ini}_t - \hat{\mathbf{P}}^{trans2}_t$ captures target-specific structural patterns and noise. Neighborhood smoothing retains meaningful target patterns while suppressing random fluctuations. The final estimate $\hat{\mathbf{P}}_t = \hat{\mathbf{P}}^{trans2}_t + \hat{\mathbf{P}}^{res}_t$ effectively combines transferred knowledge (shared patterns) with target-specific information.
>
> **Empirical support of the effectiveness of the debiasing step**:  We conducted an ablation study comparing our method with its variant without the debiasing step. As shown in **Figure C2 in the Appendix**,  GTRANS consistently outperforms its ablated variants GTRANS-NonDebias. This highlights the importance of the debiasing step.
>
> [1] Li, S., Cai, T. T., & Li, H. (2022). Transfer learning for high-dimensional linear regression: Prediction, estimation and minimax optimality. Journal of the Royal Statistical Society Series B: Statistical Methodology, 84(1), 149-173.
>
> > Comment 4:  What makes the debiasing step adaptive?
>
> Response: The debiasing step is adaptive because it's applied only when GW distance $d > \delta$, where threshold $\delta$ is chosen via cross-validation to minimize prediction error, making it data-driven based on measured source-target similarity.
>
> > Comment 5: What's `source-target domain shift'?
>
> Response:  Following transfer learning literature [1], "domain shift" refers to differences in data distribution between source and target domains. In our work, we define source-target domain shift as the GW distance between true latent edge probabilities $\mathbf{P}_s$ and $\mathbf{P}_t$, estimated using GW distance between $\hat{\mathbf{P}}_s^{\text{ini}}$ and $\hat{\mathbf{P}}_t^{\text{ini}}$ (Section 3.2.2, lines 194).
>
> [1] Wilson, G., & Cook, D. J. (2020). A survey of unsupervised deep domain adaptation. ACM Transactions on Intelligent Systems and Technology (TIST), 11(5), 1-46.
>
> > Comment 6: Why called residual matrix'?
>
> Response: We use "residual matrix" by analogy to regression analysis, where residuals represent the portion of observed outcomes not explained by the fitted model. In our setting, $\hat{\mathbf{P}}_t^{ini} - \hat{\mathbf{P}}_t^{trans2}$ captures the component of the target's initial estimator not explained by the transferred source estimator, isolating target-specific structural patterns and sampling noise not captured by transfer.
>
> > Comment 7: Lines 189-190: why additional smoothing?
>
> Response: The additional neighborhood smoothing step after the projection is implemented because the transfer step, while structurally aligning the source and target, may introduce local discontinuities or irregularities, especially when the alignment matrix aggregates nodes with different local structures. As a result, the transferred estimator might inherit sharp transitions or noise not present in the original smooth source graphon.  **Appendix C.3** provides empirical justification: our ablation study shows GTRANS-NonSmooth (omitting additional smoothing) achieves higher MSE than the full GTRANS method in most cases, demonstrating that additional smoothing improves estimation accuracy.
>
> > Comment 8: Line 229- where is the lower bound on \epsilon?
>
> Response:  The lower bound on $\epsilon$ is specified in Theorem 4.1, where we require that the penalty parameter satisfies $|\pi^*|_\infty \leq \frac{\epsilon}{C_1 | \mathbf{P}_s \otimes \mathbf{P}_t |{\mathrm{op}}}$ for some $C_1 > 2$. This condition ensures the necessary local convexity for the analysis to hold. For completeness, we will revise line 229 to explicitly reference Theorem 4.1 and clarify that the lower bound on $\epsilon$ is given there.
>
> > Comment 9:  Theorem 4.1: Does this cover the specific case where the source graph is completely uninformative about the target graph?
>
> Response: Theorem 4.1 establishes that the estimated alignment matrix approximates the population counterpart, i.e., minimizer of GW distance between  $\mathbf{P}_s$ and  $\mathbf{P}_t$, with deviation governed by the estimation error of initial estimation $\hat{\mathbf{P}}^{ini}_s$ and $\hat{\mathbf{P}}^{ini}_t$.  As long as both the source and target graphons are estimated well (e.g., via the neighborhood smoothing method, which we adopt in our work), and the parameter $\epsilon$ exceeds a certain threshold, this theorem holds, regardless of whether the source is informative or not.
>
> > Comment 10: Experiments when n_t =100, n_s=150, 200 and when n_t=150 and n_s=200, 250.
>
> Response:  Following your suggestion, we have conducted additional experiments. In **Table R5**, we compared our method (GTrans-EGW and GTrans-GW) with the Target-only method NS. The results show that GTRANS consistently outperforms NS across all cases. Additionally, as the target sample size $n_t$ increases, the relative gain from transfer learning diminishes, which is expected since more target data naturally reduces the advantage of borrowing strength from a large source.
>
> Table R5. MSE (multiplied by 100) of Graphon 1.
>  | $n_s$ | $n_t$ | GTrans-EGW | GTrans-GW | NS      | Relative Gain (%) |
>  | ---- | ---- | ----------------- | --------------- | ------------- | -------------------- |
>  | 150  | 100  | **1.40** | 1.57   | 1.83 | 23.50               |
>   | 200  | 100  | **1.39** | 1.58    | 1.83 | 24.04               |
> | 200  | 150  | **1.37** | 1.49  | 1.59 | 13.84               |
>  | 250  | 150  | **1.34** | 1.43 | 1.59  | 15.72              |
>
> > Comment 11: Lines 54-57: Are the two problems (alignment and unsupervised learning) linked?
>
> Response: These are distinct challenges. **Alignment Challenge:** Absence of known node correspondences between graphs of different sizes can cause structural patterns to be incorrectly aligned.  **Unsupervised Learning Challenge:** Unlike conventional transfer learning with labeled data, graphon estimation relies solely on observed adjacency matrices without ground-truth edge probabilities, preventing explicit loss function definition.
>
> > Comment 12: update references on graphon estimation.
>
> Response: We have conducted an empirical study comparing our method with GWB, IGNR, and SIGL. Please refer to **Table R3** in response to Reviewer aKUK. These references will be added.

---

> > ### Author Response · Authors · 2025-08-05
> > **Follow-up on our rebuttal**
> >
> > Dear Reviewer,
> >
> > Thank you again for taking your valuable time and giving constructive feedback to our paper. As the discussion period is approaching to an end, it would be very helpful if you could kindly let us know whether you have any additional queries or whether we have not clarified any of your previous queries. We are eagerly looking forward to hearing from you.
> >
> > Best,
> >
> > Authors

---

### Official Review · Reviewer_xFxn · 2025-07-02

**Clarity:** 3
**Significance:** 3
**Originality:** 3
**Rating:** 4
**Confidence:** 4

**Summary:**

This paper explores the graphon estimation issue for small graph and proposes the method to transfer the more accurate estimation of graphon from a larger similar source graph to the small target graph. Specifically, their method involves three stages being the initial estimation of the graphon for both graphs, optimizing for the alignment matrix and the final debiasing step for significant distribution shift. This paper claims to be the first work targeting small graph graphon estimation without known node correspondence and exceed the traditional baselines.

**Questions:**

**Questions:**
- Are there any criterions or measurements that suggest a good choice of source graph or have you encountered the case where the transferability is nearly impossible even with the designed debiasing process?

**Ethical Concerns:**

["NO or VERY MINOR ethics concerns only"]

**Final Justification:**

The rebuttal that clarifies my questions, I will remain my score for the paper.

**Limitations:**

Yes.

**Quality:**

3

**Strengths And Weaknesses:**

**Strengths:**
- The paper tackles a practical issue with graphon estimation for small graph and think of the novel idea to enable the transferability of graphon from a similar larger source graph.
- The method proposed is grounded with some theoretical guarantee and the design for each step is clear and reasonable. Also, the paper includes some evidence for method motivation, like the visualization of alignment matrix from adjacency-based and estimation based, as well as they discuss the strength and drawbacks of their design, e.g. comparing against GW and EGW.
- The paper itself is well-written and easy to follow

**Weaknesses:**
- The success of the small target graph estimation seems to rely a bit on whether there is a large and similar graph that can serve as the source graphon estimation. Maybe, it could be better if the authors provide more real life examples to suggest that it is very common and easy to find the similar source graphs to work on.
- It is good that we have the real dataset application being enhancing the graphon estimation used in the G-Mixup method. I'm wondering that is there any other applications and usage to showcase the benefit from more accurate small graph graphon estimation.

---

> ### Author Rebuttal · Authors · 2025-07-31
>
> > Comment 1:  The success of the small target graph estimation seems to rely a bit on whether there is a large and similar graph that can serve as the source graphon estimation. Maybe, it could be better if the authors provide more real life examples to suggest that it is very common and easy to find the similar source graphs to work on.
>
> Response: Thank you for raising this important point. In many real-world settings, it is actually quite common to find large graphs that are structurally similar to smaller ones, especially within the same application domain. **For example**, in our experiments (**Table 2**), we successfully transferred from large protein networks in D&D to smaller ones in PROTEINS-Full, and from large social or collaboration graphs (Reddit-B, COLLAB) to smaller IMDB-B or IMDB-M co-actor networks. **Another example** is in our link prediction task (**Section D.2 in Appendix**), we successfully transferred from large Wiki-Vote to  smaller four networks, including   Dolphins, Firm, Football and Karate. For your covenience, we put the link prediction results in Table A.
>
> Beyond our main experiments, here are some **potential examples**: (1) In systems biology, large gene regulatory or protein interaction networks (e.g., from STRING or BioGRID) can be used as source for smaller subnetworks in specific tissues or cell types. (2) In online social platforms, massive public graphs (from Facebook, Twitter, or Reddit) can be leveraged to inform smaller networks. (3) In neuroscience, large-scale brain connectomes can be sources for smaller regional networks. (4) In transportation research, traffic network of a large city (e.g., Chicago, NY) can be used to improve the traffic system of small urban areas.  Due to space limitations, further empirical studies on these additional datasets are beyond the scope of this work.  We believe our results provide a strong foundation, and we hope to systematically explore these broader applications in future work.
>
>
> We will revise our manuscript to include discusssions about these potential application examples.
>
>
>
> Table A: Link prediction AUC (Mean ± SD, multiplied by 100) using Wiki-Vote as the source graph ($n_s = 889$).
>
> | Dataset  | GTRANS-GW | GTRANS-EGW | NS    | USVT   | SAS      | ICE     |
> | -------- | --------------- | ---------------- | ------------- | ------------- | ------------- | ------------- |
> | Dolphins ($n_t=62$)   | 75.96 ± 8.53    | **76.26 ± 8.54**     | 70.60 ± 9.01  | 72.36 ± 10.16 | 50.66 ± 6.35  | 75.36 ± 8.40  |
> | Firm ($n_t=33$)    | **71.31 ± 12.18**   | 71.26 ± 12.06    | 66.32 ± 12.27 | 65.56 ± 12.25 | 54.90 ± 7.59  | 64.26 ± 12.20 |
> | Football($n_t=115$) | 86.64 ± 3.66    | **86.74 ± 3.72**     | 86.75 ± 3.49  | 85.32 ± 3.56  | 44.56 ± 7.38  | 82.46 ± 4.59  |
> | Karate ($n_t=34$)    | 82.47 ± 10.36   | **82.53 ± 10.46**    | 76.74 ± 12.01 | 71.86 ± 15.20 | 63.88 ± 11.15 | 80.43 ± 11.22 |
>
>
>
> > Comment 2: It is good that we have the real dataset application being enhancing the graphon estimation used in the G-Mixup method. I'm wondering that is there any other applications and usage to showcase the benefit from more accurate small graph graphon estimation.
>
>
> Response: Thank you for your encouraging comment and interest in broader applications of our method. Beyond the G-Mixup graph classification task, we have evaluated GTRANS on the **link prediction** problem, as detailed in **Appendix D.2**. Our results in **Table A** show that enhanced graphon estimation significantly improves link prediction performance compared to baselines, highlighting the method’s versatility across practical tasks.
>
>
>
> In essence, **GTRANS can enhance any application requiring graphon estimation**. For instance, researchers have utilized estimated graphons for **community detection** [1] and **graph neural network** training [2]. Due to page constraints, we defer empirical exploration of these areas to future work.
>
>
> [1] Klimm, F., Jones, N. S., & Schaub, M. T. (2022). Modularity maximization for graphons. SIAM Journal on Applied Mathematics, 82(6), 1930-1952.
>
> [2] Hu, Z., Fang, Y., & Lin, L. (2021, December). Training graph neural networks by graphon estimation. In 2021 IEEE International Conference on Big Data (Big Data) (pp. 5153-5162). IEEE.
>
>
>
> > Comment 3: Are there any criterions or measurements that suggest a good choice of source graph or have you encountered the case where the transferability is nearly impossible even with the designed debiasing process?
>
> Response:  Thank you for the valuable comment.
>
>
> **Criterion for Source Graph Selection**: As mentioned in Section 3.2.2 (line 194), we measure the source-target domain difference  using  the Gromov-Wasserstein (GW) distance between the initial graphon estimators $\hat{\mathbf{P}}^{\text{ini}}_s$ and $\hat{\mathbf{P}}^{\text{ini}}_t$, each obtained using the neighborhood smoothing method. When multiple candidate source graphs are available, we use the estimated GW distance between inital graphon estimator  as the criterion for selecting the most appropriate source graph for transfer (see Section D.1 in the appendix, lines 982-992). Our new **theoretical analysis**  (see detailed response for Comment 2 of Reviewer JN9a)  shows that $GW_2^2(\hat{\mathbf{P}}^{ini}_s, \hat{\mathbf{P}}^{ini}_t)$ **approximates** $GW_2^2(\mathbf{P}_s, \mathbf{P}_t)$, thereby justifying its use as a criterion for source selection with theoretical guarantees.
>
> **Cases Where Transferability is Challenging**: We have indeed encountered scenarios where transferability is nearly impossible due to high dissimilarity (large $d$). In our simulations (Table 1 and Table C2), cross-graphon transfers between highly dissimilar pairs (e.g., between graphon 8 and 9, with distinct patterns like gradient vs. striped structures; Figure 3) achieves similar performance as target-only neighborhood smoothing method. However, we would like to emphasize that the debiasing step in our method effectively mitigates the influence of the source graph when it is substantially dissimilar to the target domain.
>
> Thus, while GTRANS excels with similar sources, it remains safe and non-degrading even when transfer is challenging, making it a reliable tool for practitioners.

---

### Official Review · Reviewer_aKUK · 2025-07-03

**Clarity:** 4
**Significance:** 3
**Originality:** 3
**Rating:** 4
**Confidence:** 4

**Summary:**

The paper presents GTRANS, a method designed to estimate the graphon underlying a small target graph by transferring knowledge from a larger source graph with similar properties. The approach begins by estimating graphons for both the small target graph and the large source graph using neighbor smoothing. It then computes the Gromov-Wasserstein (GW) distance between these two graphons and uses the resulting alignment matrix to map the source graphon's structure onto the latent domain of the target graph. Finally, the method assesses potential domain shift by comparing the transferred graphon to the initially estimated one and smooths the residual if a significant difference is detected.

**Questions:**

Please refer to the questions mentioned in the weaknesses above.

**Ethical Concerns:**

["NO or VERY MINOR ethics concerns only"]

**Final Justification:**

The idea of transferring knowledge from a large graph to improve the graphon estimation from a smaller graph is a nice addition to the graphon estimation literature. The rebuttal from the authors included several useful additional experiments and insights.

**Limitations:**

yes.

**Paper Formatting Concerns:**

No paper formatting concerns.

**Quality:**

3

**Strengths And Weaknesses:**

Strengths:
1. The paper is well-written and easy to follow.
2. Transferring knowledge from a large graph to aid in estimating the graphon of a smaller graph is a novel and interesting idea, particularly applicable to some real-world scenarios where observed graphs are small.
3. In the experiments, the proposed method achieves better results compared to the baselines.

Weaknesses:
While the idea of transferring knowledge from a larger graph is compelling, the method assumes the large graph comes from a different but known to be similar graphon.
This raises some questions:
How do we determine which dataset's graphon is similar to the one at hand? In other words, how do we know the source is truly similar to the target? The method assumes access to an external large graph that is known to be similar to the smaller graph, but there is no formal or mathematical definition of this similarity.

As a follow-up to the above:
Why not use larger graphs from the same dataset? For example, in the G- Mixup dataset, although IMDB-B is a small dataset, it contains a few larger graphs that could be used. These might even be more beneficial than transferring from a different dataset. That is, what happens if fs=ft.

If we compare transferring from a large graph in the same dataset vs. from a large graph in a different dataset, what would the performance difference be? Intuitively, the former should yield better results. Some experiments in this direction would strengthen the paper.

Along the same lines, what if we pooled source and target graphs together and estimated the graphon jointly from multiple graphs?

The method does not compare to more recent, resolution-free graphon estimation techniques that estimate the graphon from a single graph or a set of graphs. Examples include GWB [1], IGNR [2], and SIGL [3], whereas this work relies only on simpler graphon estimation baselines.
[1] Xu, Hongteng, et al. "Learning graphons via structured gromov- wasserstein barycenters." Proceedings of the AAAI Conference on Artificial Intelligence.
[2] Xia, Xinyue, Gal Mishne, and Yusu Wang. "Implicit graphon neural representation." In Int. Conf. on Artif. Intell. and Stat.
[3] Azizpour, A., Zilberstein, N., and Segarra, S. (2025). Scalable implicit graphon learning. In Int. Conf. on Artif. Intell. and Stat.

What is the rationale behind smoothing the residual? While it’s true that the residual may capture domain shift, why should smoothing and adding it help improve the estimated graphon? Some justification or experimental evidence would be useful.

In the G-Mixup experiment, how is the source graph selected? There are many large graphs in the source dataset, and selecting one over another may significantly impact the alignment, especially given the diversity of the real-world graphs.

There are no experiments to validate the theoretical assumptions. This could potentially be addressed using synthetic datasets, where the ground truth alignment is available.

Minor comment: There is a typo in the caption of Figure 5: "iniial" should be "initial".

---

> ### Author Rebuttal · Authors · 2025-07-31
>
> We thank the reviewer for these insightful comments. We will correct typos, add the following clarifications, discussions, and experiments in the revised manuscript. Due to space constraints, only mean values are shown in the following experiments.
>
> > Comment 1: Similarity between source and target graphs:
>
> Response: We quantify source–target similarity using the Gromov-Wasserstein (GW) distance between their latent probability matrices $\mathbf{P}_s$ and $\mathbf{P}_t$ (Section 2.2). Since these are unobserved in practice, we use the GW distance $d$ between their initial estimators $\hat{\mathbf{P}}^{\text{ini}}_s$ and $\hat{\mathbf{P}}^{\text{ini}}_t$ as a surrogate (Section 3.2.2, line 194; ref. [62] of our method). The entropic regularized GW can also be used for efficiency. To avoid negative transfer, we apply debiasing when $d > \delta$, with the cutoff $\delta$ chosen via network cross-validation (ref. [32], Section 5.1, lines 321–327).
>
> When multiple candidate source graphs are available, we use the estimated GW distance between the initial graphon estimators as the criterion for selecting the most appropriate source graph for transfer (Section D.1 in Appendix, lines 982-992).  Our new **theoretical analysis**  (see detailed response for Comment 2 of Reviewer JN9a)  shows that $GW_2^2(\hat{\mathbf{P}}^{ini}_s, \hat{\mathbf{P}}^{ini}_t)$ **approximates** $GW_2^2(\mathbf{P}_s, \mathbf{P}_t)$, thereby justifying its use as a criterion for source selection with theoretical guarantees.
>
> > Comment 2: Larger graph from same dataset in G-Mixup framework:
>
> Response: First, we would like to clarify that in the G-Mixup framework, the authors estimate a **single graphon per class** by aligning and aggregating all graphs within that class, not individual graphons per graph. Specifically, for each class, they: (i) sort nodes in each graph by descending normalized degree, (ii) pad smaller adjacency matrices to match the largest graph, (iii) average the aligned matrices, (iv) estimate a graphon from this averaged matrix. This class-level graphon is meant to capture the common structural pattern of the class, not any individual graph. Our use of GTRANS in this context aims to improve the estimation accuracy of this shared graphon.
>
> Second, when the application demands per-graph graphon estimates (e.g., for individual graph prediction), intra-dataset transfer can be effective. As per your suggestion, we conducted additional experiments on link prediction for single graphs. Taking IMDB-BINARY as target dataset, we select the smallest graph (with $>20$ edges) as the target. For intra-dataset source graph, we considered: (1) the largest graph in IMDB-BINARY, and (2) the graph within IMDB-BINARY that has a small GW distance to the target. For comparison, we also included the largest graph from a different dataset as an inter-dataset source. Graphons were estimated using GTRANS-GW and baselines (NS, ICE, SAS, USVT). Link prediction was performed by the same procedure as described in **Appendix D.2**.
>
> As we can see from **Table R1**,  the best results are achieved by choosing the IMDB-B source graph with the smallest GW distance to the target, even if it is not the largest. Specifically, transfer from the most structurally similar IMDB-B graph yields the highest link prediction accuracy (AUC 0.96), outperforming both the largest IMDB-B source and all cross-dataset sources. This demonstrates that **transfer is most effective when the source and target are structurally similar** (small GW distance).
>
> Table R1: Link prediction AUC results on target graphs with various transfer sources.
> |Source|Target|$n_s$|$n_t$|GW|GTRANS-GW|NS|ICE|SAS|USVT|
> |:-----------------:|:------:|:------:|:---------:|:-------------:|:-------------:|:---------:|:---------:|:---------:|:---------:|
> |IMDB-B (largest)|IMDB-B |136|12|0.39|0.94|0.91|0.79|0.59|0.75|
> |IMDB-B (gw smaller)| IMDB-B |84|12|0.31| **0.96**|0.91|0.79|0.59|0.75|
> |Reddit-B|IMDB-B|41|12|0.40|0.94|0.91|0.79|0.59|0.75|
> |COLLAB|IMDB-B|191|12|0.43|0.94|0.91|0.79|0.59|0.75|
> |IMDB-M|IMDB-B|89|12|0.40|0.95|0.91|0.79|0.59|0.75|
>
> > Comment 3: Joint estimation by pooling:
>
> Response:  To address your concern, we conducted additional experiments comparing our method to two alternatives: (1) **Pooled Estimation**: pooling source and target graphs together, estimating a graphon using the G-Mixup procedure with neighborhood smoothing (NS) (see our response of your comment 2), and (2) **Pooled-then-Transfer**: using the pooled estimator as the initial source estimate $\hat{P}^{\text{ini}}_s$ in our transfer framework, followed by the GTRANS procedure (denoted PGTRANS-GW and PGTRANS-EGW). Following Section 5.2, we evaluated all methods within the G-Mixup framework for graph classification.
>
> As shown in **Table R2**, GTRANS-GW/EGW consistently **outperforms pooled estimators**.  Naive pooling obscures domain-specific features and introduces negative transfer; While Pooled-then-Transfer partially recovers through post-pooling adaptation, it cannot correct initial information dilution. In contrast, GTRANS maintains domain separation until adaptation, leveraging source information more effectively.
>
> Table R2: Graph classification accuracy (\%) across three target datasets.
> |Source|Target|GTRANS-GW|GTRANS-EGW|Pooled NS|PGTRANS-GW|PGTRANS-EGW|
> |:----:|:----:|:-------:|:--------:|:-------:|:--------:|:---------:|
> |Reddit-B|IMDB-B|76.30|**76.80**|74.30|74.65|75.15|
> |COLLAB|IMDB-B|76.25|**77.50**|74.15|73.90|74.75|
> |Reddit-B|IMDB-M|49.10|**51.27**|46.50|48.10|48.67|
> |COLLAB|IMDB-M|**50.47**|50.23|47.70|49.60|48.97|
> |D\&D|Proteins|**69.33**|68.52|66.59|67.80|67.53|
>
> > Comment 4: Comparison with recent methods.
>
> Response: We conducted additional experiments comparing our method with GWB, IGNR, and SIGL in the graph classification task. As shown in Table R3, our proposed transfer learning method outperforms these three non-transfer learning methods.
>
> Table R3: Graph classification accuracy (\%).
> |Source|Target|GTRANS-GW|GTRANS-EGW|GWB|IGNR|SIGL|
> |:----:|:----:|:--------:|:---------:|:---:|:---:|:----:|
> |Reddit-B|IMDB-B|76.30|**76.80**|75.30|74.35|73.50|
> |COLLAB|IMDB-B|76.25|**77.50**|75.30|74.35|73.50|
> |Reddit-B|IMDB-M|49.10|**51.27**|47.70|47.50|49.13|
> |COLLAB|IMDB-M|**50.47**|50.23|47.70|47.50|49.13|
> |D\&D|PROTEINS|**69.33**|68.52|63.45|65.87|67.13|
>
> >Comment 5: Why smoothing residual?
>
> Response: The residual matrix $\mathbf{R}_t = \hat{\mathbf{P}}^{\text{ini}}_t - \hat{\mathbf{P}}^{\text{trans2}}_t$ captures (a) target-specific structural patterns not explained by the transferred estimator and (b) random noise due to small sample size of the target graph. To retain meaningful structure while suppressing noise, we apply NS to the residual, without which it simply reduces to the target-only NS estimator. Our results consistently show GTRANS outperforming NS, demonstrated by lower MSE (**Figure 4 and Table 1**) and higher classification accuracy (**Table 2**), highlighting the critical role of residual smoothing.
>
> **Rationale for adding the smoothed residual in the debiasing step**:  When GW distance is large ($d > \delta$), transferred estimator $\hat{\mathbf{P}}^{\text{trans2}}_t$ alone may cause negative transfer. However, transfer remains useful as global properties like smoothness are often shared. Debiasing adaptively combines source structure with target-specific corrections via smoothed residuals. For example, graphons 7 & 8 (Figure 3) share smooth diagonal connectivity; the transferred estimator captures this smoothness while debiasing recovers sharper local patterns (Figure 5c).
>
> >Comment 6: Source selection in G-Mixup:
>
> Response: In the G-Mixup framework, a single graphon is estimated per class by aggregating and aligning all graphs within that class, rather than estimating graphons for individual graphs (see our response to your Comment 2 for details). As a result, **there is no issue of selecting a specific source graph** within the source dataset.
>
> Further, transfer is performed at the class level: for each target class, we estimate a class-level graphon. To identify a suitable source for transfer, we estimate initial class-level graphons for all source and target classes using neighborhood smoothing. We then calculate pairwise Gromov-Wasserstein distances between these initial estimators across all class pairs, to select the source class closest to the target (see Section D.1 of Appendix).
>
> >Comment 7: Validating theoretical assumptions:
>
> Response:  Theorem 4.1 relies on two key conditions: (i) $\epsilon > C_1|\pi^\star|_\infty |\mathbf{P}_s \otimes \mathbf{P}_t|{\rm op}$ for some $C_1 > 2$, and (ii) the estimation errors for $\mathbf{P}_s$ and $\mathbf{P}_t$ must be sufficiently small.
>
> While the **first condition** involves unobservable population-level quantities and cannot be directly verified, our simulations suggest it holds for a broad range of $\epsilon$. The following **Table E** reports computed lower bounds on feasible $\epsilon$ values (with $C_1 = 2.01$) for Graphons 1–3 (defined in Section E.1), each scaled to have operator norm 1, with $n_s = 100$ and $n_t = 50$. Any $\epsilon$ above the reported thresholds satisfies the condition, including $\epsilon = 0.01$ used in our real data analysis.
>
> **The second condition is well-supported by established graphon estimation literature**, as it requires $\hat{\mathbf{P}}_s^{\text{ini}}$ and $\hat{\mathbf{P}}_t^{\text{ini}}$ to be consistent. For example, Ref. [62] shows that NS-based estimators achieve rates of $\sqrt{\log n_s / n_s}$ and $\sqrt{\log n_t / n_t}$ for the source and target, respectively.
>
> Table E: Lower bound of $\epsilon$ for various choices of source and target graphons.
>
> |Source|Target 1|Target 2|Target 3|
> |:----:|:-------:|:-------:|:-------:|
> |Source 1|5.5e-4|5e-4|6e-4|
> |Source 2|4e-4|4.5e-4|4e-4|
> |Source 3|5e-4|4.5e-4|5e-4|

---

> > ### Comment · Reviewer_aKUK · 2025-08-05
> >
> > I thank the authors for their detailed responses. These have improved my understanding and appreciation for the paper. I will raise my score accordingly.

---

> > > ### Author Response · Authors · 2025-08-05
> > > **Thank you**
> > >
> > > Thank you for your thoughtful engagement and for considering our responses. We truly appreciate your precious time. We're glad our clarifications were helpful and are grateful for your raised score!

---

### Decision · Program_Chairs · 2025-09-17

**Decision:**

Accept (poster)

**Comment:**

The paper presents GTRANS, a method designed to estimate the graphon matrix underlying a small target graph by transferring knowledge from a larger source graph with similar properties. The approach begins by estimating graphons for both the small target graph and the large source graph using neighbor smoothing. It then computes the Gromov-Wasserstein (GW) distance between these two graphons and uses the resulting alignment matrix to map the source graphon's structure onto the latent domain of the target graph. Finally, the method assesses potential domain shift by comparing the transferred graphon to the initially estimated one and smooths the residual if a significant difference is detected. The paper gives theoretical guarantees and illustrations using experiments.

Strengths

S1 Three reviewers comment that the paper is well-written and easy to follow, and each step is clearly explained.

S2 The idea of estimating the graphon for a small network using a large network is novel and interesting.

S3 The proposed method is grounded in some theoretical guarantees.

S4 The results of the proposed method are better than the baselines.

S5 The paper clearly states the assumption that the graph is a small subgraph of a larger source graph with related structural similarities. Although this may limit its applicability, it is crucial to mention this in the paper.


Weaknesses

W1. The paper assumes that the small graph is similar to the large graph, but it does not provide a method to corroborate this assumption.

W2. The paper considers that the graph has a small number of nodes. Unfortunately, the definition of small is unclear throughout the paper, and it could differ among domains. There are no theoretical results that establish sample settings (n_s,n_t) and assumptions on the underlying target and source graphons.

W3. The theoretical guarantees are not empirically proven. Although this verification is challenging due to the unknown true distribution, synthetic data could be used as an alternative.

W4. The experiment and conclusions could have benefited from the use of a larger graph on the same dataset. For example, comparing the results using a large graph from the same small network, against a large one from a different network, or even combining both of them.

W5. Other baselines could have been included, such as resolution-free graphon estimation techniques that estimate the graphon from a single graph or a set of graphs.

W6. The proposed method is computationally expensive because of the Gromov-Wasserstein distance. Even though the use of an entropic regulariser alleviates the computational complexity, the theoretical results seem to be based on the original Gromov-Wasserstein distance.

Most authors consider the paper easy to read, featuring a novel problem and solution. Also, the theoretical guarantees, despite some concerns, support the paper's acceptance. Besides the theoretical concerns, most reviewers would have expected a more helpful methodology to figure out when to apply this model, and how to choose a similar, larger graph.

Regarding the discussion, the authors included new results comparing against other baselines, new datasets, and defined some similarity measures for comparison between graphs. All the reviews expressed gratitude for the responses, with some of them showing an increase in their final score.